# Alterations in promoter interaction landscape and transcriptional network underlying metabolic adaptation to diet

Yufeng Qin[1], Sara A. Grimm[2], John D. Roberts[1], Kaliopi Chrysovergis[1] & Paul A. Wade [1✉]

Metabolic adaptation to nutritional state requires alterations in gene expression in key tissues. Here, we investigated chromatin interaction dynamics, as well as alterations in cis-regulatory loci and transcriptional network in a mouse model system. Chronic consumption of a diet high in saturated fat, when compared to a diet high in carbohydrate, led to dramatic reprogramming of the liver transcriptional network. Long-range interaction of promoters with distal regulatory loci, monitored by promoter capture Hi-C, was regulated by metabolic status in distinct fashion depending on diet. Adaptation to a lipid-rich diet, mediated largely by nuclear receptors including Hnf4α, relied on activation of preformed enhancer/promoter loops. Adaptation to carbohydrate-rich diet led to activation of preformed loops and to de novo formation of new promoter/enhancer interactions. These results suggest that adaptation to nutritional changes and metabolic stress occurs through both de novo and pre-existing chromatin interactions which respond differently to metabolic signals.

[1] Eukaryotic Transcriptional Regulation Group, Epigenetics and Stem Cell Biology Laboratory, National Institute of Environmental Health Sciences, Research Triangle Park, NC 27709, USA. [2] Integrative Bioinformatics Group, Epigenetics and Stem Cell Biology Laboratory, National Institute of Environmental Health Sciences, Research Triangle Park, NC 27709, USA. ✉email: wadep2@niehs.nih.gov

Animals have evolved to cope with fluctuations in food availability and nutritional content of food that can have major consequences on health. The responses to nutrient quality and availability range on a continuum from complete lack of nutrients to overabundance. These physiologic states have differing homeostatic requirements ranging from the mobilization of storage depots that provide energy and critical biomolecules to the biosynthesis of molecules mediating deposition of energy and metabolic precursors, with accompanying alterations in gene expression programs to facilitate biological responses. Such metabolic adaptation can lead, in the extreme, to the development and progression of obesity and related diseases including non-alcoholic fatty liver disease (NAFLD)[1,2]. With the spread of the Western lifestyle, including consumption of an energy rich diet high in fat, obesity and NAFLD are an emerging health concern[3]. Under conditions of calorie excess, the liver adapts to overnutrition through dynamic reprogramming of the transcriptome, one of many adaptations necessary to maintain metabolic homeostasis. Gene regulatory mechanisms mediating liver metabolic adaptation to excessive calorie intake are incompletely understood. Detailed analysis of the role(s) of promoter–enhancer interaction dynamics may provide mechanistic insights into the progression from nutrient-induced metabolic adaptation to disease.

Within nuclei, chromosomes are organized at multiple levels, from nucleosomal building blocks to more complex structures. Current models suggest that domain organization can reflect genome accessibility as well as regulatory contacts with promoters that modulate gene expression[4,5]. These epigenomic events are cell-type specific and relative stable, but still have the flexibility to be fine-tuned in response to environmental cues[6,7]. Numerous studies indicate that diet can reprogram the epigenome and gene expression by affecting enhancers[8–10]. However, enhancers are usually located long distances away from their target genes, which makes it difficult to accurately assign specific enhancers to their target genes and delineate the details of gene regulation. Hi-C was recently used to identify genome-wide chromatin interactions and provide unique insights into higher-order chromatin organization[11]. Although unbiased, Hi-C requires very deep sequencing to characterize promoter–enhancer interactions, which are predicted to play important roles in gene regulation[12]. Promoter capture Hi-C (PCHi-C) was developed to facilitate identification of promoter-anchored chromatin interactions at high resolution[12,13], enabling characterization of both stable and stimulus-responsive promoter–enhancer loops[14]. To date, limited data are available describing the dynamics of promoter–enhancer interactions during metabolic adaptation to changes in diet and progression to overt physiologic changes leading to disease and how these processes impact gene expression.

Given this dearth of information, a deeper understanding of the how the liver epigenome responds to diet, obesity, and the dysfunctional metabolic and physiologic state associated with obesity should provide insights into adaptation and disease. Here, we employ a long-standing model of diet-induced obesity in C57BL/6J mice. Male animals are placed on defined diets matched in protein content that differ in carbohydrate content, lipid content, and energy content. C57BL/6J animals on the high energy content, lipid-rich diet regimen typically recapitulate aspects of metabolic syndrome in humans including insulin resistance, impaired glucose tolerance, dyslipidemia, and increased triacylglycerols in liver[15–21]. Hi-C and PCHi-C in combination with ChIP-seq and RNA-seq are performed on liver tissue from these animals to investigate the dynamics of promoter-anchored chromatin interactions during diet-induced obesity. We annotate genome-wide promoter-interacting regions in liver under different physiologic conditions and analyze how these chromatin interactions regulate metabolism. Furthermore, analysis of factor recruitment at activated enhancers

indicates that Hnf4α may play a central role in orchestrating dynamic changes in chromatin organization and gene regulation in response to metabolic cues. These findings provide insights into how chromatin organization contributes to gene regulation during metabolic adaptation and disease.

## Results

**Chronic obesity reprograms the liver transcriptome.** To model the gene regulatory events associated with metabolic adaptation to chronic obesity and NAFLD, we performed diet studies on male C57BL/6 mice (Fig. 1a). Animals at 5 weeks of age were acclimated to the NIEHS facility on a normal chow diet (NIH-31) and subsequently (5 animals per group) were fed well-characterized diets in which the predominant energy source is carbohydrate (70% of caloric content, Research Diets D12450B) or lipid (60% of caloric content, Research Diets D12492) for 20 weeks (details of diet composition are provided in Supplementary Data 1). Animals on the lipid-rich diet became markedly obese (two-tailed $t$ test, $p < 0.0001$) over the course of the study (Fig. 1b), with overall weight after 20 weeks nearly twice that of animals on the carbohydrate-rich diet[10]. Consistent with other studies, the obese mice had poor glucose and insulin tolerance (Fig. 1c, d) as well as a significant increase in plasma insulin and leptin levels (Fig. 1e, f), suggesting that their metabolism was dysfunctional.

To determine how obesity affects systemic metabolism, mice fed either LD or CD were placed in TSE phenoMaster cages, and their oxygen consumption ($VO_2$), carbon dioxide production ($VCO_2$) and energy expenditure (EE) were measured. As shown in Fig. 1g–l, average EE, $VO_2$ and $VCO_2$ values decreased in the obese (LD) group compared with CD group. Significantly higher plasma concentrations of total cholesterol, high-density lipoprotein (HDL), low-density lipoprotein (LDL), aspartate transaminase (AST), and alanine transaminase (ALT) were found in the obese group, which indicated a systemic response to obesity that includes NAFLD (Fig. 1m, n). Adipose tissue and liver mass were all significantly higher in LD group (Fig. 1o). As shown in Fig. 1p–r, mice fed LD showed an increase in adipocyte size when compared with CD. Liver sections from study animals also demonstrated altered histology in obese animals on the lipid-rich diet, consistent with the development of NAFLD (Fig. 1s).

To address the transcriptional response to obesity, we prepared RNA from liver from study animals (5 animals per group) for RNA-seq. Differentially expressed gene (DEG) analysis identified 2066 genes upregulated in obese animals on the lipid-rich diet and 1663 genes upregulated on the carbohydrate-rich diet ($N = 25,494$, FDR < 0.05, fold change > 1.2; Fig. 2a, Supplementary Data 1).

As anticipated, analysis of DEGs indicated enrichment for lipid metabolic processes in animals consuming a lipid-rich diet. In animals on a carbohydrate-rich diet, pathways were enriched in homeostasis including regulation of gene transcription (Fig. 2b, c). In chronic obesity and NAFLD, lipid metabolism is perturbed, characterized by downregulation of genes mediating de novo lipogenesis (DNL) and upregulation of genes involved in fatty acid oxidation (FAO). We examined expression of genes in these two critical metabolic pathways in our data, observing that lipid-rich diet (and associated chronic obesity and adaptation to fatty liver) led to a marked suppression of de novo lipogenesis as compared with the carbohydrate-rich diet (Fig. 2d). Hallmark genes[22] activated by sterol regulatory element binding protein 1c (SREBP-1c) including elongation of long-chain fatty acids family member 6 (*Elovl6*), fatty acid synthase (*Fasn*), and stearoyl-CoA desaturase (*Scd1*) were downregulated in animals on lipid-rich as compared with animals on carbohydrate-rich diet. We also observed upregulation of hallmark genes integral to other aspects

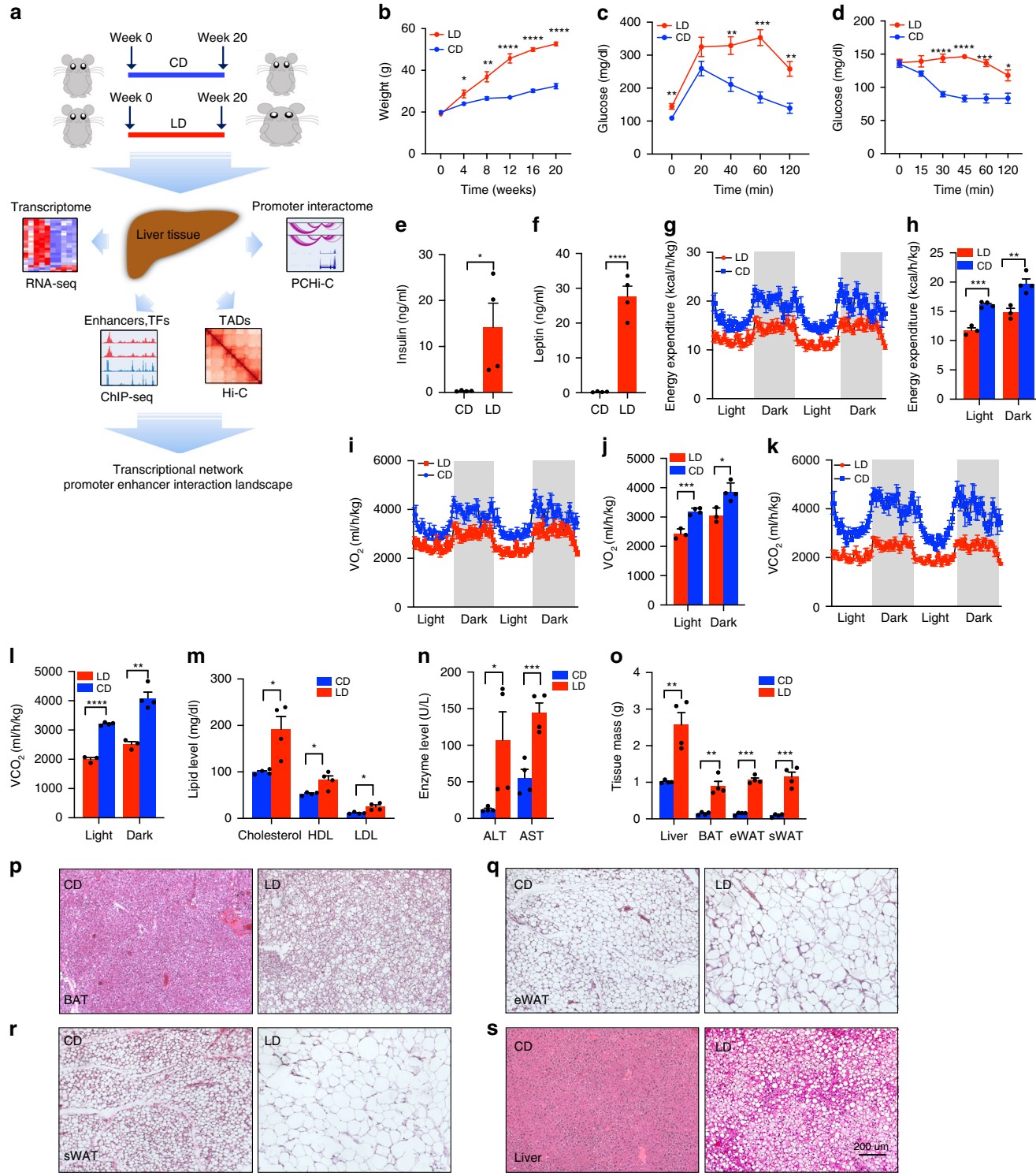

**Fig. 1 Effects of diet on metabolic parameters. a** Schematic overview of the study. **b** Body weight (mean and SEM) of study animals ($n = 5$ per group) after 20 weeks of carbohydrate-rich diet (CD) or lipid-rich diet (LD). **c**, **d** IPGTT and IPITT tests in mice fed the LD or CD ($n = 5$ per group). **e**, **f** Fasting plasma insulin and leptin level in mice fed the LD or CD ($n = 4$ per group). **g–l** Metabolic cage analysis during light and dark phase of energy expenditure (EE), oxygen consumption rate ($VO_2$), carbon dioxide production ($VCO_2$) ($n = 3$-4 per group). **m**, **n** Fasting plasma lipids and enzymes in mice fed the LD or CD. **o** Liver, brown adipose tissue (BAT), subcutaneous adipose tissue (sWAT), and epididymal adipose tissue (eWAT) mass in mice fed the LD or CD. **p–s** Hematoxylin and eosin (H&E) staining of adipose tissue and liver (×40 magnification; scale bar indicates 200 μm). Significance is indicated (two-tailed $t$ test). *$p < 0.05$, **$p < 0.01$, ***$p < 0.001$. ****$p < 0.0001$.

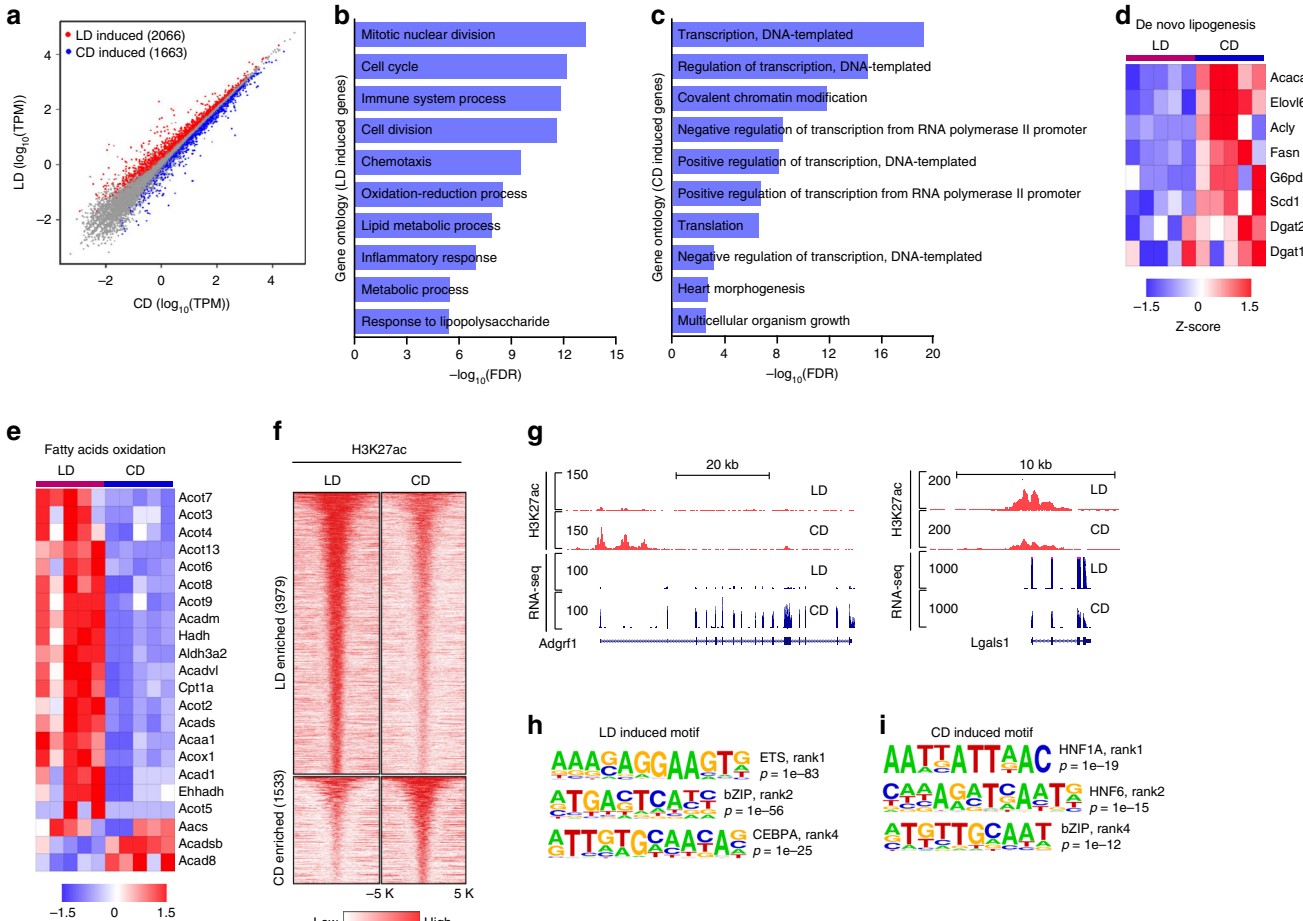

**Fig. 2 Chronic obesity reprograms the liver transcriptome and enhancer network. a** Scatter plot of RNA-seq data in mice on CD or LD. Upregulated transcripts are those higher in LD (LD induced). Downregulated transcripts are those higher in CD (CD induced). **b, c** Pathway analysis of LD/CD induced genes. **d, e** Heatmap of key genes involved in de novo lipogenesis or in fatty acid oxidation. Each column contains data from an individual study animal, each row represents a gene. **f** Heatmap of H3K27ac differentially enriched regions in LD and CD ($n = 3$ per group). Each row in the heatmap represents an individual differentially acetylated locus. LD induced loci represent loci with higher acetylation status in lipid-rich diet. CD induced loci represent loci with higher acetylation status in CD condition. **g** Example of an H3K27ac differentially enriched region. Left panel depicts the *Adgrf1* locus which has higher acetylation and transcript levels on CD. Right panel depicts the *Lgals* locus which has higher acetylation and transcript levels on LD. **h, i** Motifs enriched in LD or CD induced H3K27ac differentially enriched regions. ∗∗∗∗$p < 0.0001$.

of fatty acid metabolism in obese animals (Fig. 2e). The type one acyl-CoA thioesterases, *Acot2, Acot3, Acot4, Acot5,* and *Acot6,* which are clustered on mouse chromosome 12 and known to be regulatory targets of peroxisome proliferator-activated receptor alpha (PPAR-α), were upregulated[23]. Likewise, the type two acyl-CoA thioesterases *Acot7, Acot8, Acot9,* and *Acot13* are upregulated in obese animals. These enzymes catalyze the hydrolysis of the thioester linkage between CoA and fatty acids, leading to the generation of free fatty acid and CoA in mitochondria, peroxisomes, and the cytosol[24]. Their upregulation in dietary conditions where intracellular fatty acid concentration is elevated reflects participation in fatty acid oxidation in mitochondria and peroxisomes as well as potential roles in signaling pathways dependent on unconjugated fatty acids[25]. Also upregulated in obese animals are genes classically associated with beta oxidation of fatty acids for energy production such as acyl-CoA dehydrogenases (*Acadm, Acads*, and *Acadvl*), enoyl-CoA hydratase (*Ehhadh*), and hydroxyacyl-CoA dehydrogenase (*Hadh*) as well as the mitochondrial carnitine-dependent lipid transporter, *Cpt1*. These differences in the transcriptome are reflective of the vastly different metabolic states and regulatory networks active in

livers of animals on lipid-rich versus carbohydrate-rich diets. We selected for validation a subset of DEGs, those involved in the DNL and FAO pathways (Supplementary Fig. 1a).

**Chronic obesity rewires the enhancer network in liver.** In order to understand how adaptation to chronic obesity affects enhancer activity, we profiled genomic H3K27ac (3 animals per group) by chromatin immunoprecipitation sequencing (ChIP-seq) in liver. H3K27ac peaks were called by SICER and showed a significant overlap (~73–89%) across biological replicates, regardless of group (Supplementary Data 2a). Analysis with Diffbind revealed about 5000 loci (from a total of 50,205 peak loci) with significantly ($N = 5512$, FDR < 0.05, fold change > 1.5) differential enriched signal induced by diet (Fig. 2f, Supplementary Data 2b). A depiction of differentially enriched regions is presented in Fig. 2g. The top eight differentially enriched regions were validated by ChIP-qPCR (Supplementary Fig. 2a). As expected, these differentially enriched loci were mostly located far from TSS, and they were classified as enhancers (Supplementary Fig. 2b, c). GO and MSigDB analysis indicated that lipid-rich diet-induced

regions were enriched with genes related to immune function, including immune system process, leukocyte activation and regulation of immune response (Supplementary Fig. 2d, f), which is also consistent with the pathological process of NAFLD[26]. Carbohydrate-rich diet-induced regions were enriched with genes related to metabolic process, including small molecule metabolic process and carboxylic acid metabolic process (Supplementary Fig. 2e, g).

Enhancer regions usually harbor transcription factors, which bind cognate *cis*-acting DNA sequences and enable selective gene expression and regulation. To this end, we used HOMER to determine which transcription factor binding motifs were present in differentially enriched loci. In the lipid-rich diet enriched H3K27ac regions, the top enriched motifs corresponded to the known consensus binding sequences for ETS (ETS1, EHF), bZIP (FOSL2, JUN-AP1, and ATF3), and C/EBP (C/EBPA, C/EBPB, and C/EBPE) family transcription factors (Fig. 2h). ETS/AP1 binding sequences are prototypical RAS-responsive elements, which regulate cell proliferation and differentiation in response to a variety of growth factors and cytokines[27,28]. Lipid-rich diet and obesity is known to cause lipid accumulation in hepatic cells, leading to induction of stress signals and activation of ATF3, an adaptive-response gene[29,30]. The transcription factors in the C/EBP family have been shown to regulate lipogenesis[31] and motif analysis of lipid-rich diet enriched H3K27ac regions suggested that obesity and NAFLD-induced enhancer activity and gene transcription is likely regulated by transcription factors such as ETS or C/EBP family transcription factors. At carbohydrate-rich diet enriched loci, we also found that motifs for the bZIP family transcription factors and nuclear receptor (HNF1, HNF6) families were highly enriched (Fig. 2i). The HNF family transcription factors are involved in regulating the complex gene networks of lipid and carbohydrate metabolism[32]. For this family, the ratio of carbohydrate/fat was reported to change their DNA binding as well as local histone acetylation[33].

**Topologically associated domains are not altered by obesity.** The biological interplay between promoters and distal regulatory loci is complicated, with enhancers frequently located 10's to 100's of kilobases from transcription start sites. Unambiguous determination of functional relationships between putative enhancers and promoters is facilitated by analysis of chromatin conformation. To obtain detailed insight into the chromatin interactions in our system, we performed in situ Hi-C in livers from animals (2 animals per group) on carbohydrate-rich and lipid-rich diets[34]. Tissues were collected at the same time of day from each mouse to avoid circadian-induced chromatin organization events[35]. We sequenced samples to an average depth of 1 billion reads (Supplementary Data 3a) and generated high-resolution contact maps with a bin resolution of 50 and 10 kb (Fig. 3a, b) through HICExplorer[36]. The correlation of contact matrix between lipid-rich and carbohydrate-rich diets is 0.98 and 0.78 in the 50- and 10-kb bins, respectively (Supplementary Fig. 3a).

In order to understand the differences in chromatin interaction patterns based on diet, physiology, and metabolism, we segregated chromatin into A/B compartments using principal component analysis[11]. As expected, high correlation between regimens indicated diet and obesity, with all its associated physiologic and metabolic alterations, did not affect the patterning of A and B compartments (Supplementary Fig. 3b). Topologically associated domains (TADs) are proposed to reflect local folding event in the chromatin fiber wherein chromatin contacts occur more frequently. We hypothesized that TADs in liver would be stable across metabolic and physiologic change as they likely reflect

a more fundamental organizational unit of chromosomes. To test our hypothesis, we called TADs by HICExplorer; both conditions had a similar number of TADs with an average size of ~400 kb (Fig. 3c, d, Supplementary Data 3b). We compared TAD boundaries between the two groups. As expected, ~90% of the boundaries colocalized across condition (Fig. 3e), indicating that higher-order chromatin organization remains unchanged during obesity and NAFLD. These findings are largely consistent with other reports that TADs and their boundaries are largely conserved[37].

**Promoter chromatin interactions in metabolic adaptation.** In our Hi-C data, we found 34,982 significant chromatin interactions in liver from animals on lipid-rich diet and 37,185 from animals on carbohydrate-rich diet (Supplementary Fig. 3c). Chromatin interactions linking enhancers to gene promoters are considered to have regulatory roles[38]. Among our identified interactions, 28,682 and 30,502 chromatin interactions overlapped with gene promoters (TSS +/−2 kb). To obtain a high-resolution view of promoter-anchored chromatin interactions in liver and their changes upon adaptation to obesity, we performed promoter capture Hi-C (PCHi-C) (2 animals per group). Hi-C libraries were hybridized with 39,021 biotinylated probes targeting 25,747 promoters[12] and each library was sequenced to an average depth of ~1 billion reads. After QC by HICUP[39], we obtained an average 400 million unique and valid di-tags per replicate in each group, respectively (Supplementary Data 3a). High quality PCHi-C data in our study (Supplementary Fig. 4) reveals significant enrichment of promoter interactions compared with in situ Hi-C (Fig. 4a, b). The scores of chromatin loops called by CHiCAGO indicated the robustness and confidence of interactions[40]. Using the CHiCAGO interaction score (cutoff = 5), we identified around 170,000 interactions in each diet sample, which is six times more than that identified in Hi-C. After filtering out *trans*-interactions and interactions that span more than 1 Mb, 149,798 and 151,203 *cis* interactions were retained (Fig. 4c). A large portion (54%) of interactions was shared across physiologic condition, suggesting many promoter interactions are stable even at high resolution (Fig. 4d). Almost 80% of the interactions span <500 kb with a mean size of 284 kb (Fig. 4e). Approximately 13% of the interactions were between two promoters (Fig. 4f), indicating a role for promoters to act as regulatory elements for distal genes.

To understand the potential regulatory roles of the promoter interactions on gene expression, we correlated promoter interactions with gene expression. Generally, an increased number of promoter interactions was positively correlated with gene expression, supporting the hypothesis of a regulatory role for these promoter interactions[14,38] (Fig. 4g). We also observed that promoters for unexpressed genes were also enriched for interactions, which suggested that certain promoter interactions are preformed rather than induced by enhancer activation. Furthermore, these interactions significantly overlapped with *cis*-regulatory regions (DNase I hypersensitive sites, gene promoters, enhancers and CpG islands) defined using mouse ENCODE data[41] (Fig. 4h). The presence of H3K27ac in promoters and enhancers is associated with gene transcription, thus, we also tested for the intersection of H3K27ac enrichment with promoter-interacting regions (PIRs). Notably, those PIRs with H3K27ac enrichment were significantly associated with increased gene expression, supporting models stipulating that promoter contacts with active enhancers upregulate gene expression (Fig. 4i). These data established an atlas of promoter interactions in liver under differing dietary regimens including adaptation to obesity and metabolic stress.

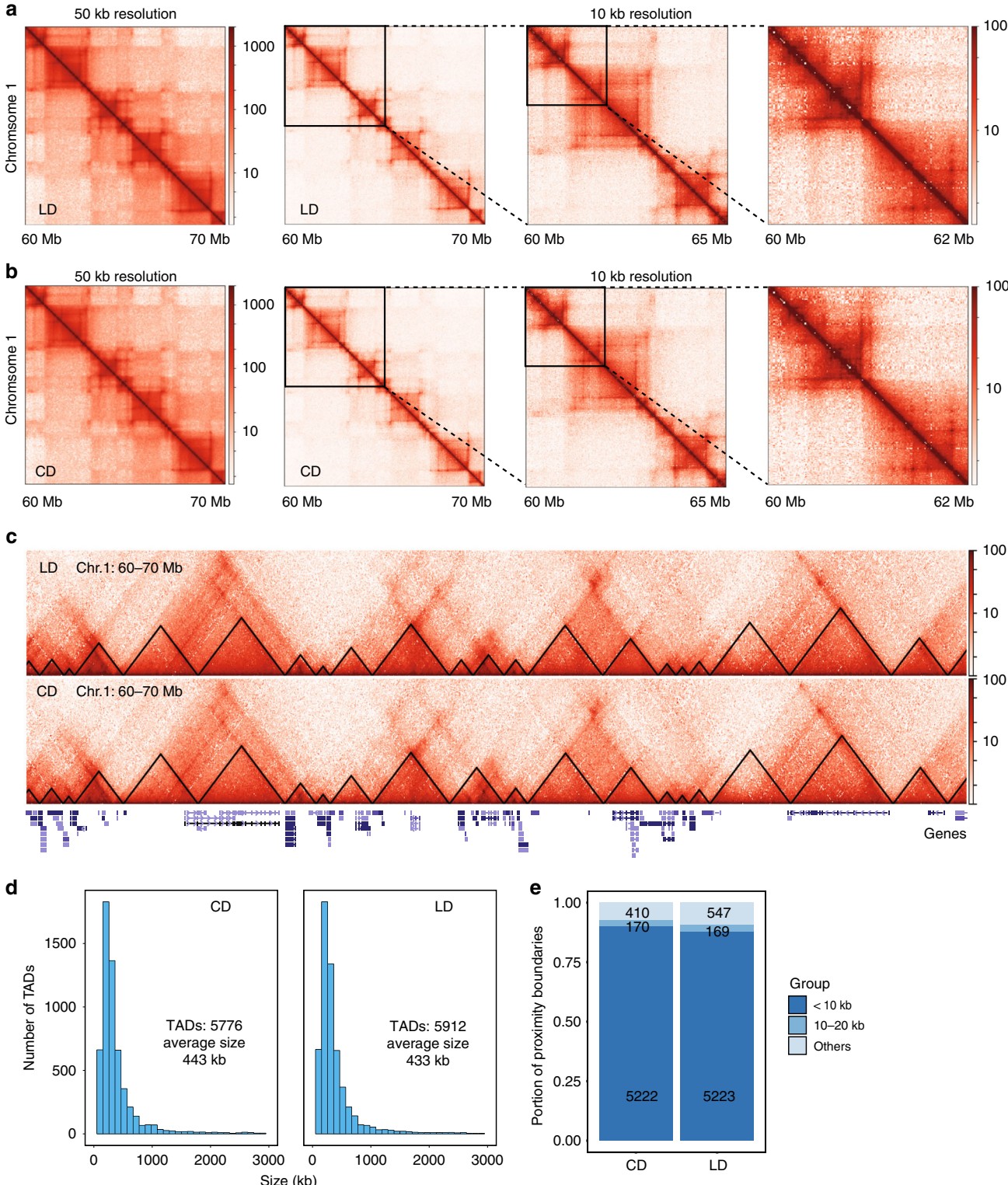

**Fig. 3 High order chromatin organization in liver during chronic obesity. a**, **b** Hi-C contact matrix depicting normalized contact frequencies in carbohydrate-rich diet (CD, Panel **a**) or lipid-rich diet (LD, Panel **b**) group at 50 and 10 kb bins ($n = 2$ per group). Examples at Chr1: 60–70 Mb. **c** Examples of TADs (Black triangle represents the TAD) and genes in Chr1: 60–70 Mb in LD and CD group. **d** Histograms of distribution of sizes of the TADs in CD and LD group. **e** Percentage of proximity TAD boundaries in CD and LD group.

**Altered promoter contacts at differentially expressed genes.** To understand whether metabolic adaptation to diet altered promoter looping events in liver, we used edgeR[38] to identify differences in chromatin interaction patterns. We identified 1962 rewired promoter interactions ($N = 195691$, $p < 0.001$); 705 were increased in frequency in lipid-rich diet and 1257 were more frequent in animals on carbohydrate-rich diet (Fig. 5a, b, Supplementary Data 3c). The interactions that were altered

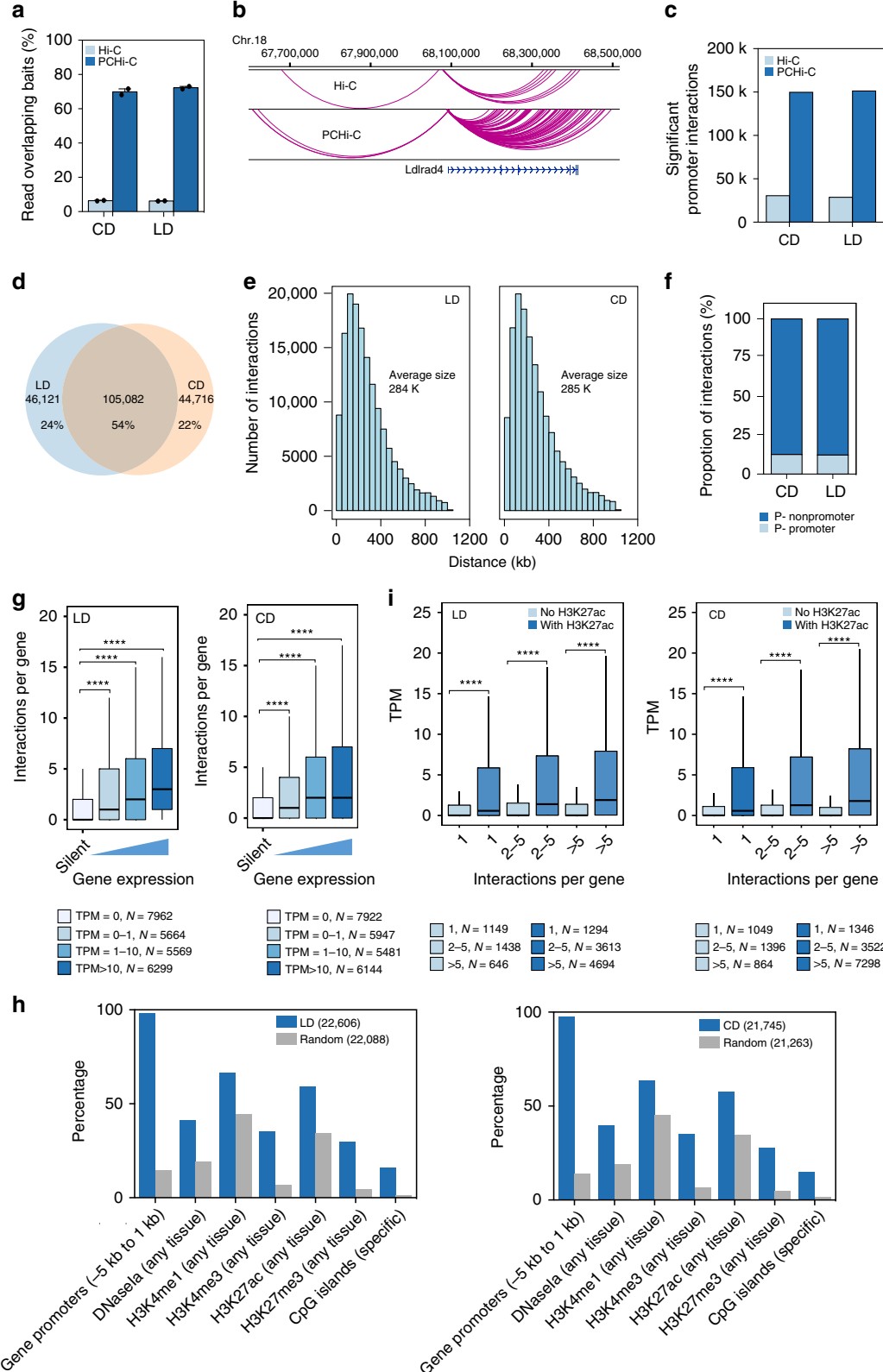

overlapped with 109 and 228 DEGs, respectively. Gene Ontology (GO) analysis indicated DEGs with carbohydrate-rich diet-induced promoter interactions were enriched in metabolism-related pathways, including small molecular metabolic process, lipid metabolism, and carboxylic acid metabolic process (Fig. 5c). DEGs with lipid-rich diet-induced promoter interactions were enriched in cell motility, regulation of transmembrane transport,

and positive regulation of cell communication (Fig. 5d). This set of pathways indicates the potential role of promoter interactions in a broad spectrum of cellular metabolic events.

We next asked whether rewiring promoter interactions contributes to altered gene expression. Compared with the DEGs associated with static loops, we found that only the promoter/enhancer interactions responsive to carbohydrate-rich diet

**Fig. 4 Promoter-anchored chromatin interactions regulating gene expression changes in metabolic adaptation to diet. a** Percent of Hi-C reads overlapping the capture Hi-C baits are shown for comparison (mean and SEM, $n = 2$ per group). **b** Example of promoter interactions identified at the *Ldlrad4* locus in Hi-C (upper panel) and PCHi-C (lower panel). **c** Significant interactions in Hi-C and PCHi-C. **d** Overlap of promoter interactions in carbohydrate-rich diet (CD) or lipid-rich diet (LD) group. **e** The distribution of distance between promoters and promoter-interacting regions (PIR) as a function of diet. **f** Percentage of promoter–promoter interactions and promoter non-promoter interactions. **g** Mean number of interactions per gene with different expression levels in two groups. Gene expression level was divided into four groups: TPM = 0, 0–1, 1–10 and >10. Box denotes 25th to 75th percentile, horizontal bar is median, whiskers extend to 1.5*IQR beyond the box, outliers are omitted. Significance is indicated (Mann–Whitney U test). **h** The percentages of interactions (TPM = 0) or matched control regions overlapping with *cis*-regulatory regions defined using mouse ENCODE data in LD and CD groups. **i** Gene expression level with different interactions with/without H3K27ac in two groups. Box denotes 25th to 75th percentile, horizontal bar is median, whiskers extend to 1.5*IQR beyond the box, outliers are omitted. Significance is indicated (Mann–Whitney U Test). ∗∗∗∗$p < 0.0001$.

significantly associated with increases in gene expression (Mann–Whitney U test, $p < 0.0001$) (Fig. 5e). Taken together, our findings demonstrate that rewiring of promoter interactions contributes to transcriptional regulation of metabolic genes and may be diet dependent.

**Preformed promoter–enhancer interactions**. An enhancer usually regulates a distal gene through physically connecting to the gene's promoter. To understand whether the activity status (using H3K27 acetylation as a proxy) of the enhancers identified as interacting with a given promoter in our capture Hi-C data reflected gene expression, we examined the overlap of the distal end of promoter interactions with H3K27ac differentially enriched loci, regardless of whether the number of contacts changed with condition. We discovered 4449 promoter–enhancer interactions in which H3K27ac was differentially enriched in lipid-rich diet and 2003 cases where this mark was enriched in carbohydrate-rich diet (Fig. 5f, g, Supplementary Data 3c). The promoter–enhancer interactions with H3K27ac enrichment that were altered overlapped with 616 and 317 DEGs, respectively. Next, we considered how gene expression changes as a function of the altered activity of enhancers. As shown in Fig. 5h, DEGs where the promoter interacts with an acetylated enhancer were upregulated, regardless of diet (Mann–Whitney U test, $p < 0.0001$). We note that the number of promoter–enhancer loops in which activation status of the enhancer changes (6452) in a preformed loop greatly exceeds the number of new loops formed following a change in conditions (1962). These findings suggest that activation of enhancers already interacting with a promoter represents an important mode of distal gene regulation during metabolic stress (Fig. 5f).

**Transcriptome analysis links Hnf4α to metabolism**. To evaluate which transcription factor binding events were involved in gene expression changes, we performed motif analysis in the gene promoters. Using HOMER, we observed enrichment of several TF motifs at the DEG promoters (Fig. 6a), including Hnf4α. HNF4α, which is a signal responsive transcription factor, has an important role in regulating lipid metabolism and is activated by lipid ligands[42]. We previously found that Hnf4α serves as a modulator to regulate the epigenome and gene expression during diet-induced obesity in the colon[10]. Since Hnf4α itself was differentially expressed in our system, we first validated its expression by qPCR and immunoblotting (Fig. 6b). We observed downregulation of Hnf4α at the protein and mRNA levels in obese animals. As the lipid-rich diet used in this study contains ~3% by weight C18 polyunsaturated fatty acids including linoleic acid (USDA, National Nutrient Database for Standard Reference https://ndb.nal.usda.gov/ndb/foods/) a known ligand for HNF4α[43], down-regulation of HNF4α protein is potentially consistent with the increased turnover noted with other nuclear receptors upon ligand binding[44]. We do not at this time understand downregulation of HNF4α mRNA. Next, we performed ChIP-seq to understand how

Hnf4α localization responds to obesity (2 animals per group). Peak calling was done by HOMER and resulted in 45,830 peaks in carbohydrate-rich diet and lipid-rich diet. Most of the peaks were located (within 500 kb) near a TSS. We compared the differentially enriched Hnf4α binding regions across condition ($N = 4384$, FDR < 0.05, fold change > 1.5) and identified 3222 peaks more enriched in lipid-rich versus carbohydrate-rich diet and 1161 peaks with the opposite enrichment (Fig. 6c, Supplementary Data 4a). The identification of more Hnf4α enriched peaks in obese animals on the lipid-rich diet is consistent with the activation of Hnf4α by lipid ligands, similar to the increase DNA binding activity of other nuclear receptors in the presence of ligand[45]. We also observed that Hnf4α binding was higher at sites that were enriched for H3K27ac, and lower at sites that were marked by lower levels of H3K27ac (Fig. 6d). These data indicated that activated Hnf4α was localized at activated promoter or enhancer loci. Examples of Hnf4α peaks specific to lipid-rich and carbohydrate-rich diet-induced genes are shown in Fig. 6e, f.

Chromatin interactions can bring distal regulatory elements into close physical proximity of target gene promoters to regulate gene expression. We next leveraged our PCHi-C and Hnf4α ChIP-seq data to determine whether Hnf4α binding is accompanied by dynamic promoter interaction events. We observed that interactions at the distal end of promoter interactions overlapped with Hnf4α binding (Chi square test, $p < 0.0001$) (Fig. 6g), which suggests that Hnf4α binds to sites of promoter interactions. We next considered Hnf4α binding at genes whose promoters had interactions with changes in occupancy (Fig. 6h). In agreement with previous findings, the DEGs that showed gains in Hnf4α peaks at the distal end of sites of promoter interactions also showed upregulated gene expression (Mann–Whitney U test, $p < 0.0001$, Fig. 6i). To extend our analysis and understand the relationship between differential Hnf4α binding and gene expression, we also assigned the condition-enriched Hnf4α binding sites to the nearest DEGs (TSS+/−50 kb). We found that DEGs with condition-induced Hnf4α binding had significantly higher expression than DEGs with unchanged Hnf4α binding (Mann–Whitney U test, $p < 0.0001$, Fig. 6j).

**Hnf4α binding with additional transcription factors**. Although we identified around 4000 Hnf4α binding sites that were changed, this number only represents about 10% of the genome-wide binding sites for Hnf4α. Of all Hnf4α binding sites, 79% are co-localized with loci enriched for H3K27ac, which is marker of active promoters and enhancers (Fig. 7a). These data suggest that most of the identified Hnf4α-bound regions in liver are active elements or open chromatin that exist in both conditions and are not readily changed. Since such a large percentage of Hnf4α binding sites are stable, we hypothesized that Hnf4α may interact functionally with other transcription factors during metabolic stress. To test our hypothesis, we used HOMER[46] to determine which transcription factor binding motifs were present in these Hnf4α bound activated regions. As expected, among 36,125

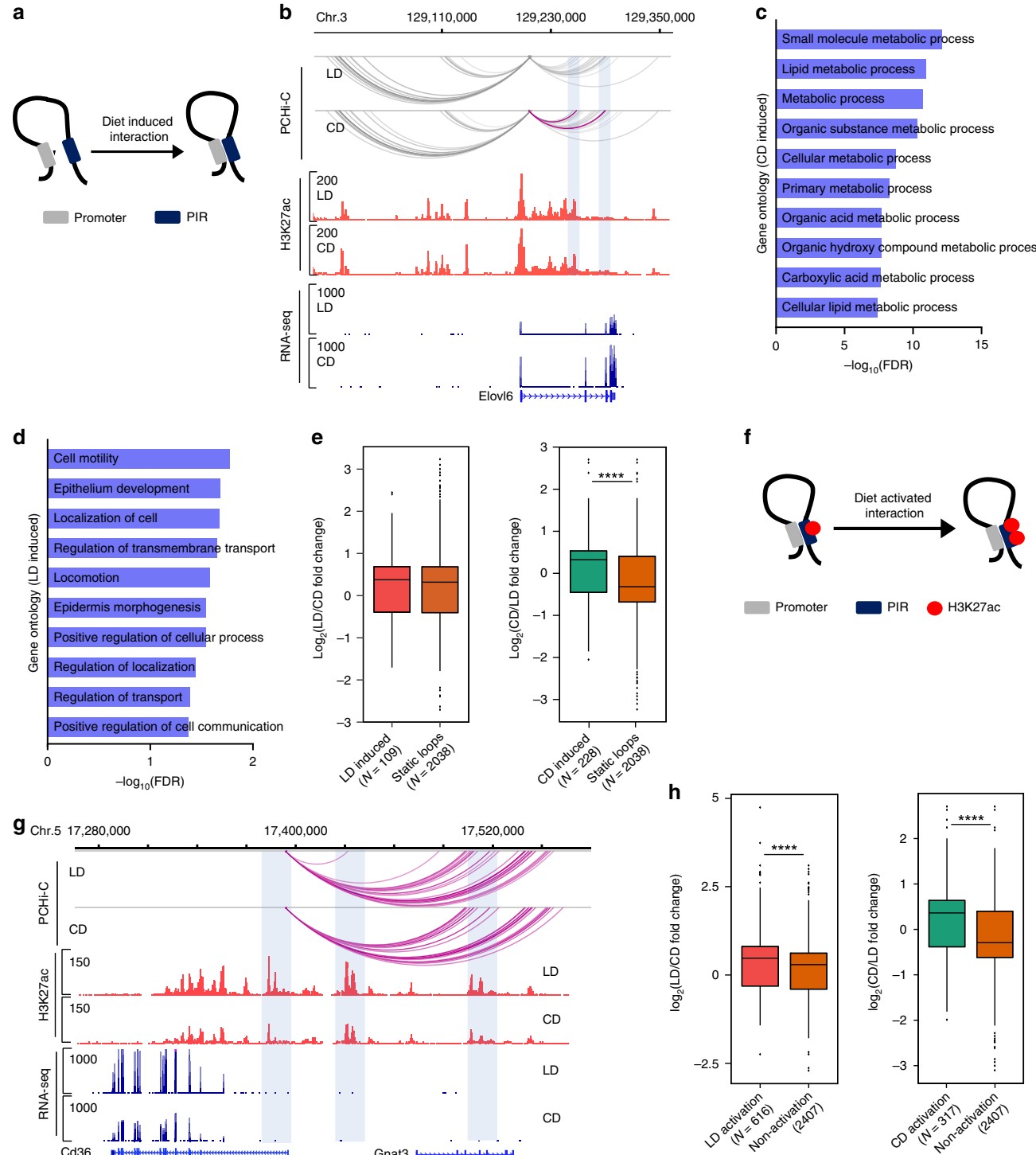

**Fig. 5 Promoter interactions rewired by diet. a** A schematic of diet-induced interactions. **b** Example of interactions with increased in frequency under carbohydrate-rich diet (CD). Interactions with CHiCAGO score > 5 were in gray color and red color indicated significantly increased interaction identified by EdgeR. **c**, **d** Gene ontology analysis of rewired interactions in CD and lipid-rich diet (LD). **e** Box and whisker plots showing changed gene expression as a function of diet-induced promoter interactions. Box denotes 25th to 75th percentile, horizontal bar is median, whiskers extend to 1.5*IQR beyond the box. **f** A schematic of diet-induced activation at promoter–enhancer interactions. **g** Genome browser view of transcript abundance (lower panels), H3K27ac (middle panels), and promoter interactions at the *CD36/Gnat3* locus. Interactions with CHiCAGO score > 5 were in red color. **h** Box and whisker plots showing altered gene expression as a function of diet at loci with activated enhancers. Box denotes 25th to 75th percentile, horizontal bar is median, whiskers extend to 1.5*IQR beyond the box. Significance is indicated (Mann–Whitney U test). ∗∗∗∗$p < 0.0001$.

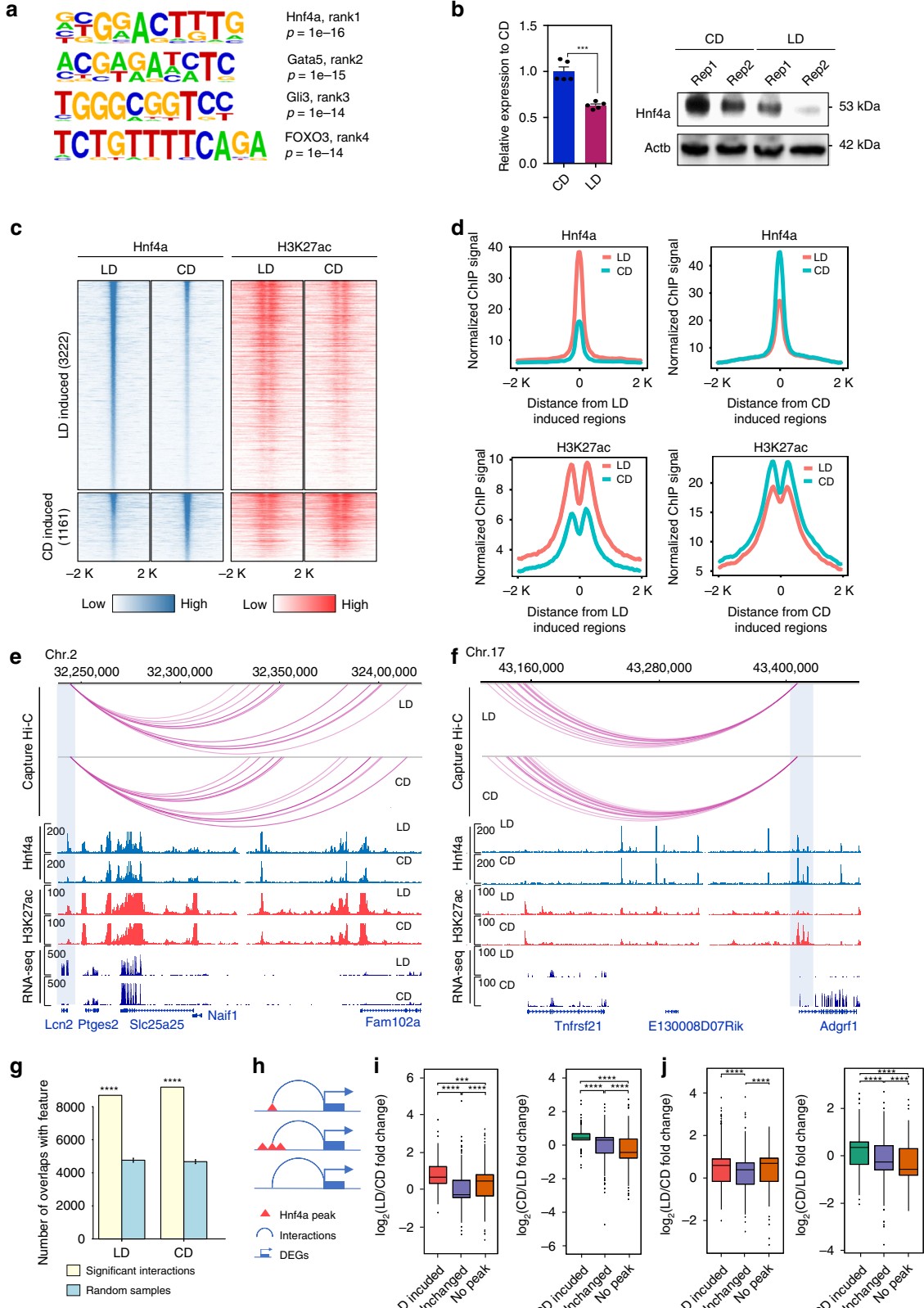

Hnf4α-bound activated regions, the most significantly enriched consensus motif is that for HNF4α. Interestingly, the second most common motif is that for the basic leucine zipper domain (bZIP) family of transcription factors, which includes the C/EBP family (Fig. 7b). C/EBPs are a family of six CCAAT/enhancer binding proteins, among which C/EBPα and C/EBPβ are most widely

expressed and studied[47]. Compared with C/EBPβ, C/EBPα is more abundant and its expression changes in our RNA-seq data (Supplementary Data 1). Thus, we focused on investigating further the location and interactions of C/EBPα.

To validate the motif results, we performed C/EBPα ChIP-seq (2 animals per group) and found 55% of C/EBPα peaks colocalized

**Fig. 6 Transcriptome analysis links Hnf4α to regulation of metabolism genes in NAFLD. a** Motifs enriched in the promoters of differentially expressed genes. **b** Abundance of Hnf4α at the transcript (left panel) and protein levels (right panel) in carbohydrate-rich diet (CD) or lipid-rich diet (LD) group. The column graph depicts mean and SEM ($n = 5$ animals). Significance is indicated (two-tailed $t$ test). **c** Heatmap of Hnf4α differentially bound regions. Acetylation status at the same loci is indicated in the right-hand panel. **d** Metagene plot of Hnf4α and H3K27ac signal at differentially bound regions. **e, f** Example of differential binding of Hnf4α and acetylation at H3K27 at the *Lcn2* and *Adgrf1* loci. Regions with differential Hnf4α binding are indicated with blue shading in the figure. **g** Overlap of Hnf4α with the distal end of promoter/enhancer interactions compared with random samples. Significance is indicated in the figure (Chi square test). **h** Schematic overview of diet-induced Hnf4α binding at interactions. **i** Box and whisker plots showing changed gene expression as a function of diet-induced Hnf4α binding at promoter interactions. Box denotes 25th to 75th percentile, horizontal bar is median, whiskers extend to 1.5*IQR beyond the box. **j** Box and whisker plots showing changed gene expression as a function of diet-induced Hnf4α binding near a TSS ($+/-50$ kb). Box denotes 25th to 75th percentile, horizontal bar is median, whiskers extend to 1.5*IQR beyond the box. Significance (Mann–Whitney U test) is indicated in the figure. ∗∗$p < 0.001$, ∗∗∗∗$p < 0.0001$.

with Hnf4α in liver (Monte-Carlo simulation, $N = 10000$, $p < 0.0001$, Fig. 7c, d). Analysis of overlapped regions by GREAT revealed that co-bound sites were enriched in genes involved in a variety of metabolic pathways, including carboxylic acid metabolism, fatty acid metabolism, steroid metabolism, and lipid catabolism (Fig. 7e). Next, we sought to understand how C/EBPα binding changes with diet and to identify the differentially bound sites ($N = 7327$, FDR < 0.05, fold change > 1.5). Using the same cutoff criteria as for Hnf4α, we observed 7327 differentially bound sites including 5102 more enriched in lipid-rich diet and 2225 sites more enriched in carbohydrate-rich diet (Fig. 7f, Supplementary Data 4b). GO analysis suggested that C/EBPα differentially bound sites were also enriched in metabolism-related pathways regardless of condition (Fig. 7g, h). Motif analysis of C/EBPα differentially bound sites also indicated an increased presence of the Hnf4α motif in both conditions (Fig. 7i, j). To investigate whether Hnf4α interacts with C/EBPα, we also examined the Hnf4α binding signal at C/EBPα differentially bound sites (Fig. 7k). Metagene plots indicated that Hnf4α signal was higher at differentially C/EBPα enriched regions, suggesting that, at a minimum, Hnf4α is always present at these sites (Fig. 7l). One example for our model is presented in Fig. 7m where the fatty acid translocase *CD36* exhibits diet dependent alterations in C/EBPα binding and histone acetylation. To test our hypothesis, we divided the diet-induced C/EBPα binding sites into two categories: sites with Hnf4α or sites without Hnf4α. We observed that co-binding of C/EBPα with Hnf4α significantly changes the binding magnitude of C/EBPα in a condition-dependent manner (Fig. 7n). We also found promoter-interacting regions where co-binding with Hnf4α significantly increased the C/EBPα binding signals at diet-induced C/EBPα binding sites in both conditions (Fig. 7o). These data suggest that Hnf4α is frequently found at promoter-interacting regions where it frequently co-binds with C/EBPα to regulate gene expression during metabolic adaptation to diet (Fig. 8).

## Discussion

Nonalcoholic fatty liver disease (NAFLD) is one of several obesity-related diseases that is part of an emerging epidemic of metabolism-based pathologies[26]. NAFLD is characterized by fat accumulation in hepatocytes and comprises a spectrum of histopathologic abnormalities ranging from simple steatosis through steatohepatitis to fibrosis and cirrhosis[2]. In a sense, NAFLD and associated pathologies can be considered as an extreme case of the spectrum of metabolic adaptation to diet which is an evolutionary necessity for animals. Here, we demonstrate that animals placed on extreme diets (an energy rich diet high in lipid, a less caloric diet rich in carbohydrate) for chronic periods exhibit alterations at the physiologic, cellular, and molecular levels. Animals consuming excess calories from a diet rich in lipid exhibit extreme weight gain at both the whole-animal level as well as in specific tissues including liver and adipose. Physiologic measures such as fasting serum hormone levels, serum lipid levels, glucose, or insulin

tolerance, and measures of energy expenditure and respiration, are likewise aberrant. In addition, chronic consumption of a lipid-rich diet leads to visible accumulation of lipid droplets in liver, to increased size of adipocytes and to accumulation of liver enzymes in serum, indicative of hepatocyte damage. These multi-system changes reflect adaptation of metabolism and physiology of the animal to an extreme diet. Not surprisingly, the chronic consumption of extreme diet and the adaptive responses described above are accompanied by molecular alterations in hepatocytes underlying dramatic remodeling of the transcriptome. Our results define physiologic responses leading to dynamic activation of pre-existing promoter–enhancer interactions and to formation of new enhancer/promoter loops (Fig. 8a, b). In the case of physiologic and metabolic adaptation to lipid-rich diet, activation of enhancers participating in preexisting interactions with promoter chromatin predominates; the responses to a carbohydrate-rich diet are more balanced.

That the metabolic response to different diets at the level of chromatin interactions differs is potentially reflective of the underlying biology. The liver response to carbohydrates is complex, but includes insulin signaling pathways that converge on the transcriptional responses mediated by the sterol regulatory element binding protein 1c (SREBP-1c). This transcription factor is sequestered in an inactive state outside the nucleus until signaling events mediate its nuclear accumulation at target genes, including a gene set differentially expressed in this study. We anticipate that such a mode of transcriptional regulation may be facilitated by the de novo creation of enhancer/promoter loops downstream of binding by SREBP-1c. In contrast, the hepatocyte transcriptome, in response to the many alterations in physiology initiated by chronic consumption of a lipid-rich diet involved differential expression of genes regulated by a pair of nuclear receptors, peroxisome proliferator-activated receptor alpha (PPARα) and hepatocyte nuclear factor 4 alpha (HNF4α). Both PPARα and HNF4α are constitutively localized to the nucleus and are activated by ligand binding. Preferential utilization of preformed promoter/enhancer loops by ligand-activated transcription factors may be reflective of a general feature of this class of transcription factors. Ligand availability for PPARα and HNF4α is expected to be modulated by diet, and potentially by the action of genes differentially expressed in this system such as the acyl-CoA thioesterases[25]. Thus, the transcriptional outcomes of adaptation to chronic consumption of lipid-rich diet likely feed forward to impact activation of master regulatory transcription factors governing the response.

In contrast to differential promoter/enhancer loops, we observed that higher-order chromatin organization, including A/B compartments and TAD boundaries, in liver is largely unchanged, even when comparing animals with such vastly different physiologic and metabolic profiles. Long-range chromatin architecture is closely associated with transcriptional programs in a cell-type-specific manner. Although our mouse model has

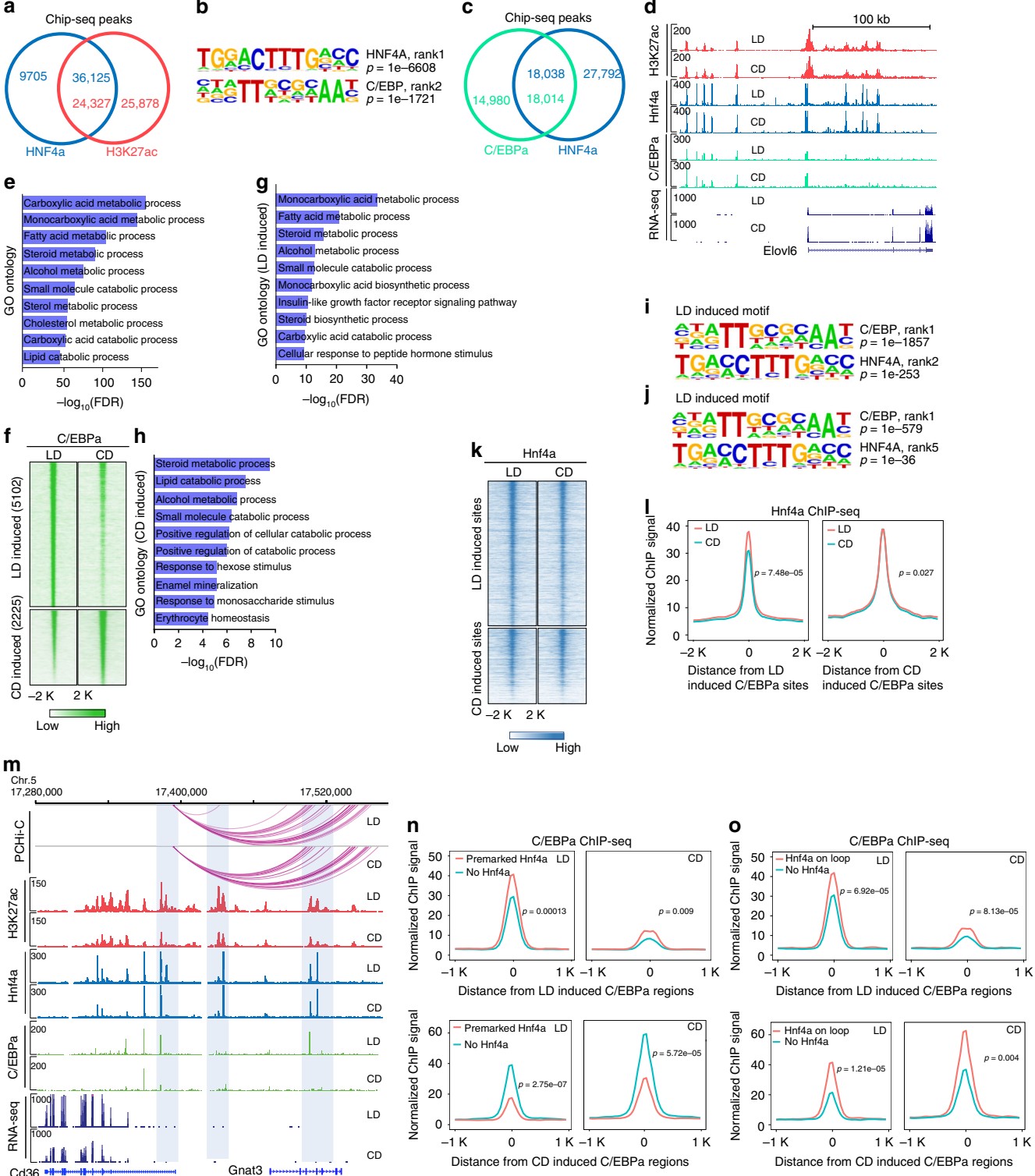

**Fig. 7 Hnf4α frequently co-binds with C/EBPα. a** Overlap between Hnf4α and H3K27ac sites, regardless of diet conditions. **b** Motifs enriched in overlapping regions of Hnf4α and H3K27ac. **c** Overlap between Hnf4α and C/EBPα binding sites, regardless of diet. **d** The genome browser shot provides an example of colocalization of Hnf4α and C/EBPα binding at the *Elovl6* locus. **e** GO analysis of Hnf4α and C/EBPα overlapping regions. **f** Heatmap of C/EBPα signal at diet-induced differentially bound regions. **g**, **h** GO analysis of C/EBPα differentially bound regions. **i**, **j** Motifs enriched at C/EBPα differentially bound regions. **k** Heatmap of Hnf4α and H3K27ac signal at C/EBPα differentially bound regions. **l** Metagene plot of Hnf4α signal at C/EBPα differentially bound regions. **m** Example of Hnf4α binding at a promoter interaction site as a bookmark for C/EBPα. **n** Metagene plot of C/EBPα at differentially bound regions with or without premarked Hnf4α. **o** Metagene plot of C/EBPα at differentially bound regions with or without premarked Hnf4α at promoter interactions. Significance (Mann–Whitney U test) is indicated in the figure.

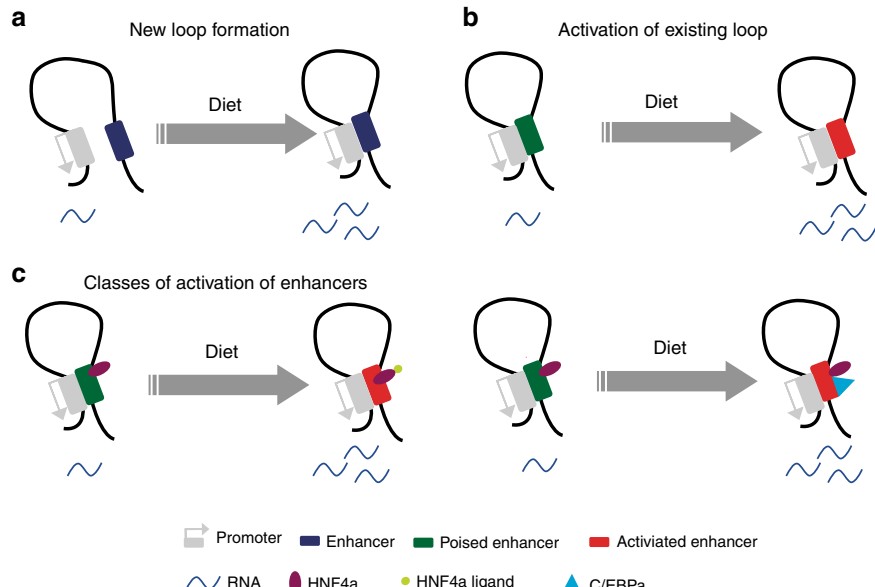

**Fig. 8 Promoter/enhancer interactions during metabolic adaptation to diet. a, b** Diet-induced differential gene expressions involves two types of promoter–enhancer interactions: de novo formation of promoter–enhancer interactions (**a**); activation of preformed promoter–enhancer interactions (**b**). **c** Enhancer activation can occur through multiple mechanisms. Nuclear receptors, including Hnf4α, can be activated by ligand binding, leading to local increases in histone acetylation and enhancer function (left panel). Alternatively, lineage-specifying transcription factors, including HNF4α, can bind constitutively at signal responsive enhancers and co-bind with other transcription factors, such as C/EBPα to regulate metabolism-related genes (right panel).

dramatic physiologic and metabolic alterations, the observed liver phenotypes are reversible on removal of the lipid-rich diet[9]. In line with this notion, chromatin structure, as revealed by topologically associated domains, appears relatively stable in NAFLD[48] despite the dramatic impact on the animal.

In hepatocytes, the balance between de novo lipogenesis (DNL) and fatty acid oxidation (FAO) maintains homeostasis. In NAFLD, cellular concentrations of fatty acids are higher than normal; only the oxidative conversion of fatty acids and a decrease in lipogenesis minimizes lipotoxicity in liver. Our gene expression data indicate that genes involved in FAO are upregulated and genes involved in DNL are downregulated (Fig. 2d, e). This finding suggested that NAFLD-induced steatosis was likely an early stage in a two-hit process of the development of NAFLD[49], in which FAO and DNL still cooperate. Gene expression change during NAFLD development is likely mediated through effects on transcription factors. Using HOMER, we found that the Hnf4α binding motif was enriched in the promoter regions of DEGs, a finding validated by Hnf4α ChIP-seq. HNF4α is an orphan nuclear receptor involved in metabolic regulation with the potential to both activate and repress transcription[50,51]. In our study, we observed that nearly 80% of Hnf4α binding sites overlapped with H3K27ac enriched loci, which suggests that Hnf4α prefers to bind at regions of open chromatin. HNF4α is primarily located in the nucleus and the protein level of Hnf4α decreased with NAFLD, suggesting that Hnf4α might be activated by its ligands and relocated in the genome to modify gene expression in NAFLD.

Combining our PCHi-C data and Hnf4α ChIP-seq, we found that Hnf4α binding was enriched at the promoter-interacting regions (Fig. 6g) involved in regulation of gene expression (Fig. 6i, j). A previous study also reported that HNF4α could recruit additional cofactors to regulate gene expression[50]. In our study, around 90% of HNF4α binding sites were stable between carbohydrate-rich diet and lipid-rich diet group, which led us to hypothesize that Hnf4α co-binding with other transcription

factors may play an important role in response to external stimuli and physiologic change. Motif analysis suggested that Hnf4α binding sites were also enriched for the binding motif of C/EBPα, another important transcription factor in metabolism. C/EBPα ChIP-seq data showed that more than half the binding sites overlapped with Hnf4α binding sites, suggesting they might work together to regulate gene expression. Importantly, we observed that more than 20% of C/EBPα binding sites were changed by different diets, which was more than the number of HNF4α sites that changed (10%). *CD36* is target gene of both HNF4α and C/EBPα, and the knockout of *CD36* protects against NAFLD during exposure to a lipid-rich diet[52]. We found that *CD36* expression was elevated under lipid-rich diet and showed an activated promoter–enhancer interaction. In the distal end of the interaction, HNF4α may function to act as a bookmark for C/EBPα, looping to the promoter of *CD36* to upregulate its expression. Collectively, these data are consistent with a model in which HNF4α premarks specific positions for C/EBPα to regulate metabolism genes (Fig. 8c).

In summary, our study presents an atlas of promoter interactions in liver during adaptation to the physiologic and metabolic outcomes of chronic exposure to carbohydrate-rich or lipid-rich diet and offers understanding about how chromatin interactions and gene expression respond to metabolic and physiologic stress. We showed that genes are regulated through promoter interaction dynamics in two different ways: activation of preformed loops and de novo generation of new loops. Furthermore, we identified significant co-binding of Hnf4α with C/EBP alpha and propose that Hnf4α may function to mark open chromatin for C/EBP. Overall, our findings provide insights into chromatin organization and gene regulation during the development of metabolism-related pathologies.

## Methods
**Mice study**. Five-week-old C57BL/6 mice were purchased from Jackson Laboratory, were acclimated to the NIEHS animal facility (on normal chow, NIH-31) for

one week and were subsequently placed on either a carbohydrate-rich diet (70% of caloric content) or a lipid-rich diet (60% of caloric content) (D12450B and D12492, respectively; Research Diets) for up to 20 weeks as previous described[10]. All animal experiments were approved by the NIEHS Institutional Animal Care and Use Committee and were performed according to the NRC Guide for the Care and Use of Laboratory Animals.

**Glucose tolerance and insulin tolerance tests**. Intraperitoneal glucose tolerance tests (IPGTTs) were performed at 19 weeks of LD and CD. Mice were fasted and glucose was injected intraperitoneally at a concentration of 2 g/kg body weight. Blood glucose was measured with a glucometer at 0, 20, 40, 60, and 120 min. Intraperitoneal insulin tolerance tests (IPITTs) were performed 1 week later. Mice were fasted and insulin was injected intraperitoneally at a dose of 0.75 IU/kg body weight. Blood glucose was measured with a glucometer at 0, 15, 30, 45, 60, and 120 min.

**Plasma parameters**. Mice were fasted overnight and blood samples were collected after 20 weeks of LD and CD. Plasma were collected and subjected to insulin measurements using the Mouse Insulin ELISA kit (ALPCO, NH, USA) and leptin measurements using the Mouse/Rat Leptin ELISA (ALPCO, NH, USA) following the manufacturer's instructions. Samples were measured in triplicate. Plasma cholesterol, HDL, LDL, ALT, and AST were measured by Clinical Pathology Laboratory in NIEHS.

**Assessment of energy metabolism**. Mice $VO_2$, $VCO_2$, and energy expenditure were measured at 19 weeks of LD and CD using the TSE phenoMaster System (TSE Systems, MO, USA) in NIEHS. TSE phenoMaster System is an open circuit indirect calorimetry system and air flow passed through each cage was controlled at ~0.4 l/min in our study. $VO_2$ and $VCO_2$ were analyzed for each individual cage every 27 min. Energy expenditure was calculated according to the TSE default setting using the following equation: total energy expenditure (kcal/h) = $3.941 \times VO_2$ (l/h) + $1.106 \times VCO_2$ (l/h). Data were collected for 3 days and normalized by weight; the first day was considered the acclimation period and was excluded from data analysis. Mice had unlimited access to food and water for the entire duration in CaloCages.

**Hi-C library preparation**. In situ Hi-C was carried out as previous described with some modifications[12,53,54]. Livers tissue was collected, homogenized and filtered with a 70-μm nylon cell strainer. Around 10 million cells were fixed with 1% formaldehyde in PBS for 10 min at room temperature and cross-linking was terminated by addition of glycine at 0.125 M for 5 min. Cells were lysed with 10 mM Tris-HCl pH 8.0, 10 mM NaCl, 0.2% NP-40 and protease inhibitors for 30 min on ice. Nuclei were resuspended in 1 × NEBuffer2 with 1% SDS and incubated at 62 °C for 10 min. After incubation, nuclei were collected and resuspended in 1 × NEBuffer2 with 1% Triton X-100. HindIII (NEB, R3104, 400 u) enzyme was added to the solution and the nuclei were digested at 37 °C overnight with agitation (750 rpm). Another 400 unit HindIII was added to the solution at 37 °C for another 1 h the next day. After digestion, 37.5 μl of 0.4 mM biotin-14-dATP (Thermo-Fisher, 19524016), 1.5 μl of 10 mM dCTP, 1.5 μl of 10 mM dGTP, 1.5 μl of 10 mM dTTP, and 8 μl of 5 U/ul Klenow (NEB, M0210) were added and rotated at 37 °C for 1 h. After dATP filled in, the solution was incubated at 65 °C for 10 min to inactive the enzyme. Ligation was performed by adding 667 μl of water, 120 μl of 10X NEB T4 DNA ligase buffer, 100 μl of 10% Triton X-100, 12 μl of 10 mg/ml BSA, and 0.5 μl of 400 U/ul T4 DNA Ligase (NEB, M0202) and incubating at 16 °C for 4 h. Reverse cross-linking was done by adding 10 μl of proteinase K (ThermoFisher, AM2546), 100 μl of 10% SDS, and 100 μl of 5 M NaCl, and incubated at 65 °C overnight. DNA was purified by phenol–chloroform extraction and 20 μg of DNA was incubated with T4 DNA polymerase (NEB, M0203) for 4 h at room temperature to remove the biotin from non-ligated ends. Then, DNA was sheared by Covaris and double size selected using AMPure XP beads as described in ref. [54]. MyOne Streptavidin T1 DynaBeads (ThermoFisher, 65601, 150 μl) were washed, resuspended in 300 μl of wash buffer (5 mM Tris-HCl, 0.5 mM EDTA, 1 M NaCl, 0.05% Tween 20) and added in 300 μl of 2× binding buffer (10 mM Tris-HCl, 1 mM EDTA, 2 M NaCl) with sheared DNA for 15 min at room temperature. The pulled down fragments were end-repaired, adenine-tailed, and ligated to adaptors (Bioo Scientific, NOVA-5144-61) on beads as described in ref. [54]. The immobilized Hi-C libraries were amplified using primers mix from NEXTflex Pre and Post Capture Kit (Bioo Scientific, NOVA-5144-61) with 12 PCR amplification cycles. The resulting libraries were sequenced on Illumina Novaseq as 50 bp paired-end.

**Promoter capture Hi-C (PCHi-C)**. Promoter Capture Hi-C (PCHi-C) was carried out using the Agilent SureSelect target enrichment (SureSelectXT Custom 3–5.9 Mb library) system. Biotinylated RNA baits were designed to the ends of HindIII restriction fragments that overlap Ensembl-annotated promoters of protein-coding, noncoding, antisense, snRNA, miRNA, and snoRNA transcripts[12]. Seven hundred and fifty nanograms of Hi-C library was captured according to the manufacturer's instructions[12]. Briefly, DNA was denatured for 5 min at 95 °C and hybridized with RNA baits at 65 °C for 24 h. MyOne Streptavidin T1 DynaBeads (ThermoFisher, 65601) were used to pull down the biotinylated DNA/RNA. After

library enrichment, the NEXTflex Pre and Post Capture Kit (Bio Scientific, NOVA-5144-61) was used to generate libraries with 10 PCR amplification cycles. The resulting libraries were sequenced on Illumina Novaseq as 50 bp paired-end.

**Hi-C data analysis**. Hi-C reads were processed by HiCUP pipeline, which aligns reads, filters artifact fragments (such as circularized reads and re-ligations), and removes duplicates[39]. The percentage of mapping efficiency, unique and validated di-tags were listed in Supplementary Data 3a. The biological replicates in each group were merged to perform the analysis. The HiCExplorer[36] was used to generate contact matrices (hicBuildMatrix for each replicate with parameters --binSize 10000 --restrictionSequence AAGCTT --inputBufferSize 100000; then hicSumMatrices to combine replicates). Normalization was performed by HiCExplorer hicCorrectMatrix (--filterThreshold -1.237 4 for LD group and --filterThreshold -1.248 4 for CD group), limiting to canonical autosomes, chrX, and chrY. TADs and TAD boundaries were defined by HiCExplorer hicFindTADs (--thresholdComparisons 0.05 --delta 0.1 --correctForMultipleTesting fdr). HiCExplorer hicPlotTADs was used for Hi-C matrix and TADs visualization. Chromatin loops were called by HiCExplorer hicFindEnrichedContacts (--method z-score --maxDepth 100000000) and filtered with z-score > = 6 and loop distance < = 1 Mb. Aggregate peak analysis plots were generated by summing the signal of pixels at loops (and local background of 100 kb) in 10 kb bins, excluding any loops within 300 kb of the matrix diagonal, then scaling by the average of the 4 corner pixels of the local background submatrix. HOMER was used to identify the A/B compartments at 50 kb bins using the following method: makeTagDirectory with parameters "-tbp 1 -restrictionSite AAGCTT -genome mm9 -removeSelfLigation", combine the two replicates per group (makeTagDirectory), then runHiCpca with parameters "-res 50000 -genome mm9".

**PCHi-C interaction analysis**. PCHi-C reads were also processed by HiCUP pipeline[39]. The percentage of mapping efficiency, unique and validated di-tags and capture efficiency were listed in Supplementary Data 3a. We used CHiCAGO[40] to call the significant interactions with score cutoff 5. All the trans-interactions and those interactions span more than 1 Mb were discarded. EdgeR was used to identify the significantly increased and decreased promoter interactions with cutoff 0.001 as described in ref. [38]. When calculating the correlation between gene expression and interaction number, TPM was divided in four categories: TPM = 0, TPM = 0–1, TPM = 1–10, and TPM > 10. To test whether H3K27ac at the distal end of the interaction, we categorized interactions according to whether at least one distal end overlapped an H3K27ac peak. To avoid the dramatic difference between non-overlap and overlap with H3K27ac, we split the genes according to the number of total associated loops, resulting no interaction, 1 interaction with/without H3K27ac, 2–5 interaction with/without H3K27ac and > 5 interaction with/without H3K27ac. Overlap of interactions with functional genomic elements was performed using EpiExplorer[41]. EpiExplorer automatically calculates control sets for user-uploaded region sets, which is done by reshuffling the genomic positions while retaining the overall number of regions and the distribution of region sizes. The selected overlap criterion is at least 10% overlap. We divided the diet-induced promoter interaction in two categories: (1) rewired interactions that were significantly different between diets which is from EdgeR ($p < 0.001$), (2) interactions that were not significantly different but with differential H3K27ac enrichment at the distal end.

**ChIP-seq and data analysis**. Livers tissue was collected, homogenized and fixed with 1% formaldehyde in PBS for 10 min at room temperature. Cross-linking was terminated by addition of glycine at 0.125 M. Fixed tissue was lysed (1% SDS, 5 mM EDTA, 50 mM Tris-HCl (pH 8.1), protease inhibitor cocktail) and sonicated using a Covaris to generate ~200 bp fragments for immunoprecipitation as previously described[10]. The 20 μg chromatin in IP buffer (1% Triton X-100, 2 mM EDTA, 150 mM NaCl, 20 mM Tris-HCl (pH 8.1)) was subjected to immunoprecipitation with 1 μg of H3K27ac (Abcam, ab4729), 5 μg Hnf4α (Abcam, ab41898) or 5 μg C/EBPα (Abcam, ab40764) antibody and incubated overnight. The samples were incubated with Dynabeads A/G beads for another 3 h. The beads were washed with the low salt (0.1% SDS, 1% Triton X-100, 2 mM EDTA, 20 mM Tris-HCl, pH 8.1, 150 mM NaCl), high salt (0.1% SDS, 1% Triton X-100, 2 mM EDTA, 20 mM Tris-HCl, pH 8.1, 500 mM NaCl), LiCl (0.25 M LiCl, 1% NP-40, 1% deoxycholate, 1 mM EDTA, 10 mM Tris.HCl, pH 8.1) and then twice with TE buffer as previously described[10]. After elution from the beads, the protein-DNA complexes were reverse crosslinked and DNA was purified by AMPure XP beads (Beckman). ChIPed DNA (1–2 ng) was prepared using NEXTflex Rapid Illumina DNA-Seq Library Prep Kit according to the manufacturer's instructions (Bioo Scientific, NOVA-5144-02). The resulting libraries were sequenced on Illumina Nextseq500 as 35 bp pair-end.

Raw reads were filtered by quality score and aligned to the mouse genome (mm9) (Supplementary Data 6). Unique aligned and deduplicated reads were used for peak calling. SICER (v1.1) was used for H3K27ac with size 200, gap size 200, fragment size 200, and FDR cutoff 0.001[55] (Supplementary Fig. 5). For Hnf4α and C/EBPα, HOMER was used to do the peak calling with " –style factor"[46]. Differentially enriched loci were identified using the DiffBind package and results were filtered with FDR < 0.05 and fold change > 1.5. Top eight differentially

enriched loci were validated by ChIP-qPCR and primers are listed in Supplementary Data 5b. The HOMER package was used to extract motifs from the differentially bound regions (–size given) against a large set of randomly selected genomic fragments of the same size. The statistical significance of the overlap between HNF4α and C/EBPα peaks was determined by Monte-Carlo simulation, where BEDtools v2.24.0 'shuffle' was used to select size-matched random peaks within enhancer regions (defined by H3K27ac peaks) followed by overlap assessment via BEDtools v2.24.0 'intersect' for 10,000 iterations.

**RNA-seq and data analysis.** RNA-sequencing was done by the NIEHS epigenomics core. Libraries were sequenced as PE-76mers on an Illumina NextSeq500. Raw read pairs were filtered to require a minimum average base quality score of 20. Filtered read pairs were mapped against the mm9 reference genome by STAR[56] with the following parameters: -- outSAMattrIHstart 0 --outFilterType BySJout --alignSJoverhangMin 8 -- outMultimapperOrder Random --limitBAMsortRAM 55000000000 (other parameters at default settings). Mapped read counts per gene were reported by Subread featureCounts (version 1.5.0-p1) with parameters -s2 -Sfr -p[57]. Differentially expressed genes (DEGs) were identified by DESeq2 (v1.10.1) using filtering thresholds of FDR < 0.05 and fold change > 1.2[58]. Normalized read counts are reported as transcripts per million (TPM). The gene models used for the RNA-seq analysis are taken from mm9 RefSeq annotations, as downloaded from the UCSC Genome Browser on September 9, 2016. Promoter motif analysis was done by HOMER with default setting. DEGs in DNL and FAO pathways were validated by qPCR or western blot and primers are listed in Supplementary Data 5a.

**Histological analysis.** Livers and adipose tissues were removed immediately after cervical dislocation and fixed in 10% buffered formalin. Paraffin sections (5.0 mm) were stained with hematoxylin and eosin (H&E) and analyzed using a microscope.

**Statistical analysis.** All statistical analyses were performed by two-tailed Student's $t$ test, Mann–Whitney U test or Chi-squared test using GraphPad Prism7 (San Diego, CA) and RStudio. The level of significance was set at $p < 0.05$ and indicated as follows: $*p < 0.05$; $**p < 0.01$; $***p < 0.001$; $****p < 0.0001$. All data were expressed as means ± SEM.

**Reporting summary.** Further information on research design is available in the Nature Research Reporting Summary linked to this article.

## Data availability

Data that support the findings of this study have been deposited in GEO with the following accession numbers: GSE124463. All other relevant data supporting the key findings of this study are available within the article and its Supplementary Information files or from the corresponding author upon reasonable request. All the code used in this study is publicly available from the cited references. The source data underlying Fig. 1b, Fig. 2b, c, Fig. 3e, Fig. 4a, c, f, Fig. 5c-e, h, Fig. 6b, g, i, j and Fig. 7e, g, h, l, n, o and Supplementary Fig. 1 and Fig. 2a, d-g are provided as a Source Datafile. A reporting summary for this Article is available as a Supplementary Information file.

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

## Acknowledgements

We thank the NIEHS Epigenomics Core for next-generation sequencing support and the NIEHS animal facility for their assistance in animal studies. This research was supported by the Intramural Research Program of the NIH, National Institute of Environmental Health Sciences (ES101965 to P.A.W.). We are grateful to the members of the Wade laboratory for critical input through the course of this work. We thank Dr. Peter Fraser for sharing his PCHi-C probe design. This paper was substantially improved by critical comments from Drs. Douglas Phanstiel and Jill Dowens.

## Author contributions

The study was designed by Y.Q. and P.A.W. Experiments were performed by Y.Q., J.D.R., and K.C. SA.G. and Y.Q. performed data analysis. The paper was written by Y.Q., S.A.G., J.D.R., and P.A.W. All authors read and approved the final paper.

## Competing interests

The authors declare no competing interests.
