## [Peer Review File · Nature Communications]

Reviewers' comments:

Reviewer #1 (Remarks to the Author):

Summary:

The authors performed an integrative genomic study to understand the transcriptional and epigenomic response in the liver when given a lipid-rich diet. Specifically, they profiled the gene expression, chromatin conformation as well as DNA modification and transcription factor binding sites in mice that were given either a lipid-rich or a carbohydrate-rich diet. While the overall chromatin structure represented by topologically associating domains (TADs) remain unaffected by the metabolic signal, the authors reported multiple cis promoter-enhancer interactions that were altered due to the lipid-rich diet. One of the most significant findings by the authors is that diet or metabolic stress influence that regulatory mechanism in the liver. A high-fat diet largely leads to activation of existing promoter-enhancer loops, whereas, a carbohydrate-rich diet encourages more de novo promoter-enhancer interactions. Furthermore, they identified a novel metabolic transcriptional regulator in the liver, Hnf4a, which preferentially binds to the promoters of diet-responsive genes. Hnf4a often co-localize with H3K27ac, a marker for active promoter or enhancer loci, indicating Hnf4a is a hallmark of active elements as well. The authors also suggested that Hnf4a is a bookmark for additional transcriptional binding in order to respond to metabolic stress. Overall, the authors present several novel insights into the regulatory mechanism of metabolic response in the liver and provide multiple pieces of evidence to support their hypotheses. However, several issues still need to be addressed.

Major comments:

1. The authors have validated 6 DEGs (Fig. S1) and 3 H3K27ac differentially enriched regions (Fig. S2A) respectively. However, it is unclear how they were selected and how many genes/loci were tested. Additional details on the validation experiments in the manuscript will help readers better understand the rationale.
2. The authors utilized edgeR to identify promoter interactions that significantly differ in frequency between the two dietary groups. The significant threshold is set at 0.001 based on a previous publication. Presumably, the authors tested the shared interactions between the two groups, 54% of 150,000 interactions=80,000 total tests. Please clarify this in the manuscript. Consequently, an alpha of 0.001 seems too liberal given the 80,000 tests. Why not apply the more conventional threshold of $FDR < 0.05$?
3. The expression of genes is stated to be significantly associated (line 251) with the presence of diet-responsive promoter interactions. However, no statistics or methods were reported in the text nor in Figure 4D and figure legends to support the claim.
4. One of the major findings of the study is that enhancer activation status change is suggested to be more prevalent than the formation of new chromatin interactions. Subsequently, a carbohydrate-rich diet is relatively more likely to drive new promoter interactions than a high-fat diet. Comparisons based on the number of significant hits are problematic as it will be greatly influenced by sample sizes, effect sizes, and statistical power. For example, if the changes of H3K27ac binding are much larger thus easier to detect than promoter interactions, then one will also observe more enhancer activation than rewired promoter interactions. One additional analysis that might be more suitable is to estimate the proportion of true null hypothesis based on the n_0 statistics from the qvalue package. Therefore, the authors can estimate what proportion of the H3K27ac bindings are true null (i.e., not different between the two groups) versus the proportion of true null for differential interactions.
5. In Figure 5F, it is shown that distribution of gene expression fold-change of genes near lipid-rich diet-induced Hnf4a peaks is shifted slightly to the left (down-regulated based on the fold change direction in Table S1). Given that Hnf4a peaks often mark active promoters and enhancers (as

seen in Figures 5D and 5E), wouldn't the expected distribution shift slightly to the right (up-regulated)?

6. The authors hypothesized that Hnf4a is a bookmark for transcription binding factor binding when under metabolic simulation. However, they observed that 90% of Hnf4a binding sites are indifferent to the dietary regimen, so this hypothesis seems counter-intuitive. Wouldn't it be more likely that Hnf4a are labeling constitutively active promoters/enhancers in liver cells given that it often binds at H3K27ac sites? It is also shown that C/EBPa differentially binding sites have an increased presence of Hnf4a; however, this is not surprising given that Hnf4a and C/EBPa often co-localize. To support the hypothesis, it will be helpful to show that C/EBPa differentially binding sites are more enriched for Hnf4a than the overall C/EBPa binding sites (background).

Minor comments:

1. The sample sizes, numbers of tests, and numbers of replicates are often not explicitly stated in the manuscript. It will significantly help the readers to understand the experiments if the information is provided in the main text.

2. Fig. 1F and 1H seem redundant given that Fig. 1E and 1G show the same results. Please consider omitting them as Figure 1 is quite dense already.

3. To calculate the correlation between gene expression (TPM) and the number of interactions, TPM values were categorized into 4 groups. Since TPM is a quantitative measurement, what is the rationale to not directly estimate Pearson's or Spearman's correlation?

4. On line 294, "We compared the differential peaks of Hnf4a...", the authors stated they identified 3222 peaks in the lipid-rich condition versus 1161 peaks in carbohydrate-rich. I believe they meant 3222 peaks displayed a higher binding affinity in a fat-rich diet, whereas, 1161 peaks had higher Hnf4a binding when given a carbohydrate-rich diet. The current sentence seems to suggest that 3222 peaks were exclusively identified in lipid-rich mice and vice versa. It will help the readers to understand the analysis if the sentence is rephrased.

Reviewer #2 (Remarks to the Author):

In this work Wade and colleagues investigate the effect of high fat diet on hepatic gene expression and chromatin landscape in mice. The authors compare livers of mice exposed to either a high fat diet (60% of caloric content is lipid) or high carbohydrate diet (70% of caloric content is carbohydrate) for 20 weeks. Using RNA-seq and ChIP-seq it is demonstrated that the two diets lead to significant differences in the transcriptome which are paralleled by differences in enhancer activity. The authors then move on to investigate the impact of the different dietary challenges on the chromatin 3D structure. Using conventional Hi-C they show that the overall chromatin organization of the liver is mostly stable across dietary condition. However, with promoter capture Hi-C, there are some interactions that change in strength, and the authors hypothesize that different diets affect that transcriptome by different mechanisms. Motif analysis of promoters of differentially expressed genes identified HNF4A as a potential regulator, but HNF4A ChIP-seq revealed that binding largely is stable across dietary conditions. The authors propose that HNF4A acts as a bookmarking factor that facilitates the binding of C/EBPa which is differentially expressed and bound to chromatin during between the two diets.

Major points:

1. The biological rationale for comparing the two extreme dietary challenges is not clear, and expect for body weight and sections of the liver there is little information on the physiology of the

mice. It is also unclear whether the high carbohydrate diet is a just low-fat diet, or whether it has high sucrose. What is more specifically the question?

2. The authors report that RNA-seq data indicates that the high fat diet is associated with increased expression of genes involved in fatty acid oxidation, while the high carbohydrate diet is associated with increased expression of genes involved in de novo lipogenesis. The transcriptome of mouse livers in the context for high fat diet have been probed multiple times (Leung et al. 2014, van der Heijden et al. 2015, Wang et al. 2016, Siersbaek et al 2017), and these findings are not novel.

3. The authors characterize the hepatic epigenome by H3K27ac ChIP-seq, and discover approximately 4500 regions differentially occupied by H3K27ac in the two dietary conditions. The data quality is questionable, since the correlation between samples within the same dietary condition is not higher the correlation between samples across diets. It would be beneficial to explore data quality further by for example PCA.

4. The authors report that regions differentially occupied by H3K27ac are enriched in the vicinity of genes related to immune function, and speculate that this reflects the increased liver inflammation associated with NAFLD. This conclusion is not particularly convincing since the regions are not stratified by dietary condition. Thus, it is not clear whether the regions enriched for H3K27ac in the HFD condition are actually associated with inflammatory genes. The authors also perform motif analysis in the differentially occupied regions, but do not comment on the potential implications of their findings. Here, it would also be beneficial to stratify the regions by the diet in which they show enrichment of binding.

5. The authors perform conventional Hi-C as well as promoter capture Hi-C but unfortunately these data are not really used in the manuscript to provide new biological insight. The authors show that the majority of A/B compartments as well as TADs do not change between the two diets. However, similar findings have been reported in several other systems, and as such it does not provide new insight. It could have be interesting to characterize the TADs or A/B compartments that do change, but unfortunately the authors did not do this.

6. The authors claim based on promoter capture Hi-C and H3K27ac ChIP-seq that HFD is associated mostly activation of enhancers that regulate genes across stable loops, while the high carbohydrate diet is associated with both enhancer activation and regulation of promoter interactions. This claim is not very convincing, since for promoter capture Hi-C, N=2 which indicates a very low statistical power to identify differential interactions in a sparse dataset. Furthermore, rigorous quality control and analysis of inter-replicate variance that may confound the number of up- and downregulated interactions are missing.

7. Using motif analysis in the promoters of differentially regulated genes, the authors identify HNF4A as a potential transcriptional regulator and map it using ChIP-seq in the two dietary conditions. Since the majority of HNF4A binding sites are stable between the dietary regimes, the authors speculate that HNF4A is not directly regulating dietary reprogramming, but serves as a bookmark for another transcription factor. Using motif analysis in HNF4A-bound enhancers, they find enrichment of the C/EBP motif and profile C/EBPa using ChIP-seq. The evidence that HNF4A bookmarks C/EBPa enhancers is based on co-occupancy analysis and lacks both statistical and experimental support.

Specific points:

- Figure 1
- o Figure 1D: The p-values should be adjusted for multiple testing.

- o Figure 1D: Are the upregulated genes in HFD associated with a different set of pathways compared to the downregulated genes?
- o Figure 1K: Are the upregulated genes in HFD associated with a different set of motifs compared to the downregulated genes? H3K27ac is a mark of the nucleosomes surrounding the nucleosome free region of enhancers. This nucleosome free region is the primary binding site of transcription factors to enhancers. Motif analysis should be redone to focus on these regions rather than the entire H3K27ac region.
 - Figure 2
- o Figure 2C/E + Figure S3B: There are clearly a significant number of regions that do change between diets. One example is in the center of the visualized regions, where two TADs merge. Are the genes that change between diets associated with these dynamic TADs or A/B compartment switches?
 - Figure 3
- o Figure 3A: Capture efficiency for Hi-C is nonsensical, and cannot be used as a gauge for quality or enrichment.
- o Figure 3B/C: Have the Hi-C libraries and PChi-C libraries been subsampled to the same number of validation reads? Were significant interactions detected by the same method?
- o Figure 3G (and manuscript lines 224-226): The authors state that even silent genes are enriched for interactions, which are present prior to enhancer activation. Can the authors comment on the number of false positives expected in their data? How many of these preformed loops connect to restriction fragment associated with enhancer marks in either diet or how many have the potential to be enhancers in different settings (e.g. in ENCODE data)?
- o Figure 3H: Related to the above question concerning false discoveries. Does this trend persist if the analysis is restricted to very high confidence interactions (Chicago score > 10)? The conclusion could be strengthened by showing that not only does gene expression associate with H3K27ac in an on/off manner, but also in a linear correlating relationship.
 - Figure 4
- o Figure 4A/B (and manuscript lines 237-241): Please clarify if these tests have been adjusted for multiple testing and what the expected level of false discoveries is? What is the magnitude of change in either direction?
- o Figure 4C: Which pathways are associated with interaction in each diet?
- o Figure 4D: How many genes associated with LD-enriched interactions is also significantly higher expressed in LD compared to CD and vice versa?
- o Manuscript lines 265-268: The authors note that the number of significantly changing interactions is less than the number of significantly changing H3K27ac sites. For these numbers to make sense, it is important to consider the statistical power of the assays. The PChi-C dataset is likely to be sparse (which increases variation) and has N=2, while H3K27ac is likely more dense and has N=3. Thus, the conclusion here is confounded by differences in statistical power, and this needs to be taken into careful consideration.
- o Manuscript lines 268-274: Along the lines above, these conclusions might also be confounded by differences in statistical power, which needs to be carefully reviewed. An analysis of replicate similarity would be necessary. Furthermore, it would be sensible to subsample the libraries to a similar size. In the full dataset, the CD, which is associated with the majority of enriched contacts, is also sequenced approximately 40 million valid read pairs deeper per replicate. However, even with these updates, it is a difficult conclusion to reach considering the low replication (which is common for these types of experiments, but a hindrance for exactly this type of conclusion).
- o Figure 4G: How many genes associated with interactions with LD-enriched H3K27ac signal is also significantly higher expressed in LD compared to CD and vice versa?
 - Figure 5
- o Figure 5A: These motif analyses should be performed in enhancers connected to the promoters of DEGs rather than the promoters of DEGs alone, and in a manner that separated out the individual diets.
- o The authors speculate that HNF4 α is activated by lipid-derived ligands in the lipid-rich diet regime, since there is an increased number of binding sites in this condition, although there is also less protein present. How is the nuclear abundance of HNF4 α affected by diet? Both in total and

relative to the amount of HNF4a present.

o Figure 5F: These analyses should be performed by using the PCHi-C data rather than associating enhancers to genes by linear distance. Furthermore, it would be beneficial to choose an analysis and plot that is easier to interpret than the cumulative distribution.

- Figure 6

o Figure 6A: The choice of control region is critical for this type of analysis, and randomized HindIII regions are a poor choice as this underestimates the background. In the literature there are examples of other ways. Probably the best way is to shuffle the promoter labels of the detected interactions, as this preserves the structure of the data set but randomizes the location of the anchors in the genome. Another possible solution is for each bait-target pair to choose the distance-matched HindIII fragment in the opposite direction of the target as a control.

o Figure 6B: Please include DEGs that do not show gain of HNF4a binding as controls.

o Figure 6E: Is this overlap significant? This can be addressed by shuffling the binding sites of each factor randomly between enhancers and evaluating the randomized overlap.

o The authors speculate that HNF4a bookmarks enhancers for C/EBP α , since it does not change in occupancy at the sites, where C/EBP α changes. The evidence present here is not sufficient to arrive at this conclusion, as there is no evidence to suggest any relationship between their binding other than co-occupancy. This claim should be backed by experimental evidence, e.g. HNF4a knockdown followed by C/EBP α ChIP(-seq).

Minor points:

- Overall

o The nomenclature of diet regime should be consistent throughout the paper. The diets are referred to as CD/LD, HFD/LFD or control/NAFLD in a mixed manner.

- Figure 1

o The understanding of the dataset could be improved by an introductory schematic figure of the experimental setup and readouts

o Figure 1C mentions up- and downregulated genes, while the manuscript (lines 106-107) mentions genes upregulated in either of the diets.

o The visual impressions of all screenshots (Figure 1F, H and J) can be improved.

- Figure 2

o The zoom-ins should be highlighted differently than dotted lines in Figure 2A/B. They are not easy to see or figure out that they demarcate regions of interest.

o For consistency of analysis, it would be beneficial to show the length distributions of interactions in the same manner for Hi-C (Figure 2D) and PCHi-C (Figure 3E).

o Figure 2E: It might be easier to interpret this figure if simply the number or percentage of total boundaries that are conserved between diets and the number of boundaries that aren't is shown.

- Figure 3

o Other papers have found similar findings as in Figure 3G. It would be nice to cite a few of these papers in the text to increase the confidence and accumulating evidence in the field that this relationship exist.

- Figure 4:

o Figure 4E: The interactions in Figure 4E do not seem to be anchored in a promoter based on the annotation show in the figure.

- Figure 5

o Figure 5B: What is Control?

o Figure 5D: There is a spelling mistake on all x-axis labels in the figure.

o Figure 5E: Layer PCHi-C on top.

o Figure 5F: The tick marks should include 0.

- Figure 6

o Manuscript lines 346-348: The sites are enriched under a specific diet, not specific.

o Figure 6K (and manuscript lines 355-357): The authors use the screenshot as an example of their model of adaptation to diets. It would be helpful to have a schematic that highlights the important regulatory connections rather than a screenshot.

Reviewers' comments:

Reviewer #1 (Remarks to the Author):

Summary:

The authors performed an integrative genomic study to understand the transcriptional and epigenomic response in the liver when given a lipid-rich diet. Specifically, they profiled the gene expression, chromatin conformation as well as DNA modification and transcription factor binding sites in mice that were given either a lipid-rich or a carbohydrate-rich diet. While the overall chromatin structure represented by topologically associating domains (TADs) remain unaffected by the metabolic signal, the authors reported multiple cis promoter-enhancer interactions that were altered due to the lipid-rich diet. One of the most significant findings by the authors is that diet or metabolic stress influence that regulatory mechanism in the liver. A high-fat diet largely leads to activation of existing promoter-enhancer loops, whereas, a carbohydrate-rich diet encourages more de novo promoter-enhancer interactions. Furthermore, they identified a novel metabolic transcriptional regulator in the liver, Hnf4a, which preferentially binds to the promoters of diet-responding genes. Hnf4a often co-localize with H3K27ac, a marker for active promoter or enhancer loci, indicating Hnf4a is a hallmark of active elements as well. The authors also suggested that Hnf4a is a bookmark for additional transcriptional binding in order to respond to metabolic stress. Overall, the authors present several novel insights into the regulatory mechanism of metabolic response in the liver and provide multiple pieces of evidence to support their hypotheses. However, several issues still need to be addressed.

Major comments:

1. The authors have validated 6 DEGs (Fig. S1) and 3 H3K27ac differentially enriched regions (Fig. S2A) respectively. However, it is unclear how they were selected and how many genes/loci were tested. Additional details on the validation experiments in the manuscript will help readers better understand the rationale.

We thank the reviewer for drawing to our attention the lack of explanation for how we selected regions/genes for validation. We have added more details in the Results to make our choices transparent to the reader and made new figure S1a and figure S2a. For DEGs, we picked eight DEGs in specific pathways - de novo lipogenesis and fatty acid oxidation - for validation. For the H3K27ac differentially enriched regions, we validated the top eight LD induced and CD induced regions. The validation results were consistent with the RNA-seq and ChIP-seq data. Figures are presented here for the convenience of the reviewers.

2. The authors utilized edgeR to identify promoter interactions that significantly differ in frequency between the two dietary groups. The significant threshold is set at 0.001 based on a previous publication. Presumably, the authors tested the shared interactions between the two groups, 54% of 150,000 interactions=80,000 total tests. Please clarify this in the manuscript. Consequently, an alpha of 0.001 seems too liberal given the 80,000 tests. Why not apply the more conventional threshold of FDR<0.05?

We apologize for not being clear in the original manuscript. The overlap in figure 3d was generated by comparing the genomic location of promoter interactions; no statistical tests were performed in the overlap analysis. For the significantly different interactions in edgeR, we tested all interactions identified in either the LD or CD groups, which was approximately 195k tests.

As this reviewer suggested, we used the q-value package to calculate the portion of true null hypotheses of the interactions. The pi0 value is 0.704 (figure a below, only presented here), indicating that 70% of the interactions aren't different between the LD and CD groups. Our cutoff ($p < 0.001$) resulted in only 1962 different interactions (~1%) and is still conservative. In a different Capture HI-C study, the authors used FDR 0.1 as cutoff for significantly different interactions and FDR 0.7 as cutoff for unchanged/stable interactions to avoid the false positive and false negative results (PMID: 28805829). We understand that multiple tests can cause false positive results. There were about 195k tests performed in EdgeR and our 0.001 cutoff was relative to q-value 0.07 (figure b below, only presented here), which is still conservative (PMID:12883005).

3. The expression of genes is stated to be significantly associated (line 251) with the present of diet-responsive promoter interactions. However, no statistics or methods were reported in the text nor in Figure 4D and figure legends to support the claim.

Thanks for this comment. We added the fold change of genes with static loops under different diet condition in new figure 4e. Compared to the genes with static loops, only the CD induced loops had significantly higher gene expressions (Mann-Whitney U test). We added the statistics in results and figure legend to figure 4e.

4. One of the major findings of the study is that enhancer activation status change is suggested to be more prevalent than the formation of new chromatin interactions. Subsequently, a carbohydrate-rich diet is relatively more likely to drive new promoter interactions than a high-fat diet. Comparisons based on the number of significant hits are problematic as it will be greatly influenced by sample sizes, effect sizes, and statistic power. For example, if the changes of H3K27ac binding are much larger thus easier to detect than promoter interactions, then one will also observe more enhancer activation than rewired promoter interactions. One additional analysis that might be more suitable is to estimate the proportion of true null hypothesis based on the n_0 statistics from the qvalue package. Therefore, the authors can estimate what proportion of the H3K27ac bindings are true null

(i.e., not different between the two groups) versus the proportion of true null for differential interactions.

Thanks for this comment. We agree that the sample size will affect the number of significant differential bindings/interactions. We estimated the proportion of true null hypotheses using q-value package as you suggested (see table and figure below, only presented here). The percentage of true null hypotheses is similar in H3K27ac binding and chromatin interactions, suggesting they have similar backgrounds. If we use the same cutoff, the number of H3K27ac differential binding sites are larger than the number of differential interactions. Using the current cutoff, we still found the total number of interactions was 4 times the number of H3K27ac binding sites, the number of differential interactions were only one third of differential H3K27ac binding sites. This suggested that the change in enhancer activation was more dramatic than the promoter interactions. We understand the reviewer's concern and modified the conclusions to better express this concept.

	Total number	Number of differential bindings / interactions	Pi0
H3K27ac	50,205	5,512	0.6766
PChI-C	195,691	1,962	0.7044

5. In Figure 5F, it is shown that distribution of gene expression fold-change of genes near

lipid-rich diet-induced Hnf4a peaks is shifted slightly to the left (down-regulated based on the fold change direction in Table S1). Given that Hnf4a peaks often mark active promoters and enhancers (as seen in Figures 5D and 5E), wouldn't the expected distribution shifts slightly to the right (up-regulated)?

Thanks for this comment. As also suggested by reviewer 2, we regenerated the figure 5f using PCHI-C data and also by assigning DEGs to differentially enriched Hnf4a binding sites using linear distance. In new figure 5i, we categorized the DEGs in 3 groups like figure 5h indicated: DEGs with loops connected to stable Hnf4a binding sites, diet induced Hnf4a binding sites, and no Hnf4a binding sites. We found that DEGs on the loops with diet induced Hnf4a binding sites had significantly higher expression than those on the loops with stable Hnf4a binding in both diet conditions. Due to the limitations of capture Hi-C, it had less power in capturing promoter enhancer interactions occurring over short (within 50 kb) distances (figure 3e). We assigned diet induced Hnf4a binding sites to the TSS +/-50kb of DEGs and also found that DEGs with diet induced Hnf4a binding had significantly higher expression than DEGs with unchanged Hnf4a binding. These data supported the idea that diet induced Hnf4a binding correlated with upregulation of gene expressions under long term adaptation.

6. The authors hypothesized that Hnf4a is a bookmark for transcription binding factor binding when under metabolic simulation. However, they observed that 90% of Hnf4a binding sites are indifferent to the dietary regimen, so this hypothesis seems counter-intuitive. Wouldn't it be more likely that Hnf4a are labeling constitutively active promoters/enhancers in liver cells given that it often binds at H3K27ac sites? It is also shown that C/EBP α differentially binding sites have an increased presence of Hnf4a; however, this is not surprising given that Hnf4a and C/EBP α often co-localize. To support the hypothesis, it will be helpful to show that C/EBP α differentially binding sites are more enriched for Hnf4a than the overall C/EBP α binding sites (background).

Thanks for this comment. We agree that Hnf4a is labeling the active enhancers in liver cells and added this in the results. Knocking out the Hnf4a in mice liver will significantly decrease the C/EBP α binding, which means Hnf4a is not only co-localized with C/EBP α , but also has functional effects on C/EBP α (PMID:23911320). The challenge to performing this experiment is that whole body knockout of Hnf4a is embryonic lethal (PMID:7958910) and mice with liver-specific knockout of Hnf4a will die around 8 weeks age (PMID:11158324). Thus, it's

hard to study the long term (20weeks) metabolic adaptation using Hnf4a knockout mice. To understand Hnf4a's role on the C/EBPa binding during long term metabolic adaptation, we took our data and divided the diet-induced C/EBPa binding sites into two categories: sites pre-marked with Hnf4a and sites without Hnf4a (in figure 6n). We saw that diet-induced C/EBPa binding sites pre-marked with Hnf4a changed the binding magnitude of C/EBPa in a diet-dependent manner (figure 6n). While considering the Hnf4a binding on the loops, we found it was different and loop connected Hnf4a increased C/EBPa binding in both diet condition (fig 6o). These data, while not a perfect experiment, support our hypothesis that Hnf4a might be functional as a bookmarker for C/EBPa during long term metabolic adaptation.

Minor comments:

1. The sample sizes, numbers of tests, and numbers of replicates are often not explicitly stated in the manuscript. It will significantly help the readers to understand the experiments if the information is provided in the main text.

We added the sample replicates and number of tests in the results and highlighted them according to your suggestions.

2. Fig. 1F and 1H seem redundant given that Fig. 1E and 1G show the same results. Please consider omitting them as Figure 1 is quite dense already.

We removed Fig1E and 1G according to your suggestions.

3. To calculate the correlation between gene expression (TPM) and the number of interactions, TPM values were categorized into 4 groups. Since TPM is a quantitative measurement, what is the rationale to not directly estimate Pearson's or Spearman's correlation?

Thanks for the reviewer's suggestion. We calculated Spearman's correlation of TPM and number of interactions under different diet conditions. In LD group (figure a below, only presented here), the rho is 0.31 and $p < 2.2 \times 10^{-16}$. In CD group (figure b below, only presented

here), rho is 0.30 and $p < 2.2 \times 10^{-16}$. The number of interactions were significantly correlated with gene expression, which supports our conclusion. Because there's a large portion of small TPM values, we categorized TPM into 4 groups with similar number of genes in each group for comparison and for better visualization (see below, figure 3g).

4. On line 294, "We compared the differential peaks of Hnf4a...", the authors stated they identified 3222 peaks in the lipid-rich condition versus 1161 peaks in carbohydrate-rich. I believe they meant 3222 peaks displayed a higher binding affinity in a fat-rich diet, whereas, 1161 peaks had higher Hnf4a binding when given a carbohydrate-rich diet. The current sentence seems to suggest that 3222 peaks were exclusively identified in lipid-rich mice and vice versa. It will help the readers to understand the analysis if the sentence is rephrased.

We rephrased the text to help readers to understand the results according to your suggestions.

Reviewer #2 (Remarks to the Author):

In this work Wade and colleagues investigate the effect of high fat diet on hepatic gene expression and chromatin landscape in mice. The authors compare livers of mice exposed to either a high fat diet (60% of caloric content is lipid) or high carbohydrate diet (70% of caloric content is carbohydrate) for 20 weeks. Using RNA-seq and ChIP-seq it is demonstrated that the two diets lead to significant differences in the transcriptome which are paralleled by differences in enhancer activity. The authors then move on to investigate the impact of the different dietary challenges on the chromatin 3D structure. Using conventional Hi-C they show that the overall chromatin organization of the liver is mostly stable across dietary condition. However, with promoter capture Hi-C, there are some interactions that change in strength, and the authors hypothesize that different diets affect that transcriptome by different mechanisms. Motif analysis of promoters of differentially expressed genes identified HNF4A as a potential regulator, but HNF4A ChIP-seq revealed that binding largely is stable across dietary conditions. The authors propose that HNF4A acts as a bookmarking factor that facilitates the binding of C/EBP α which is differently expressed and bound to chromatin during between the two diets.

Major points:

1. The biological rationale for comparing the two extreme dietary challenges is not clear, and expect for body weight and sections of the liver there is little information on the physiology of the mice. It is also unclear whether the high carbohydrate diet is a just low-fat diet, or whether it has high sucrose. What is more specifically the question?

The formula of the diet used in our study is listed below. The physiology of the mice under these diets have been studied extensively, so we didn't focus on that. The CD is widely used and usually called "low fat diet", but we think the name is not appropriate. From the formula (taken from the manufacturer's information), we can see that CD contains 34% sucrose and 30% corn starch, which is much higher than in the LD which contains less than 10% sucrose. More and more studies have indicated that a high sugar/carbohydrate diet is not good for health, while no study published to the present has focused on the long-term metabolic adaptation to this diet and the gene regulatory network under this dietary condition.

	Ingredient	CD	LD
Protein	Casein, Lactic, 30 Mesh	200.00 g	200.00 g
Protein	Cystine, L	3.00 g	3.00 g
Carbohydrate	Sucrose, Fine Granulated	354.00 g	72.80 g
Carbohydrate	Lodex 10	35.00 g	125.00 g
Carbohydrate	Starch, Corn	315.00 g	0 g
Fiber	Solka Floc, FCC200	50.00 g	50.00 g
Fat	Lard	20.00 g	245.00 g

Fat	Soybean Oil, USP	25.00 g	25.00 g
Mineral	S10026B	50.00 g	50.00 g
Vitamin	Choline Bitartrate	2.00 g	2.00 g
Vitamin	V10001C	1.00 g	1.00 g
Dye		0.05 g	0.05 g
	Total:	1055.05 g	773.85 g
Caloric Information			
Protein:		20% kcal	20% kcal
Fat:		10% kcal	60% kcal
Carbohydrate:		70% kcal	20% kcal
Energy Density:		3.82 kcal/g	5.21 kcal/g

2. The authors report that RNA-seq data indicates that the high fat diet is associated with increased expression of genes involved in fatty acid oxidation, while the high carbohydrate diet is associated with increased expression of genes involved in de novo lipogenesis. The transcriptome of mouse livers in the context for high fat diet have been probed multiple times (Leung et al. 2014, van der Heijden et al. 2015, Wang et al. 2016, Siersbaek et al 2017), and these findings are not novel.

We do not claim that transcriptome changes under different diets is a novel finding. Rather, it is how high-level chromatin organization and promoter/enhancer interactions change due to diet that is novel, as we are not aware of similar results having been published elsewhere. We have established an atlas of promoter/enhancer interactions in liver under long-term metabolic adaptation. Also, we found certain signal response factors at the promoter/enhancer loops might have different functions. Furthermore, the general concept emerging from our study, that promoter/enhancer dynamics rather than chromatin reorganization are occurring during metabolic stress, provides clues for how to proceed with a mechanistic analysis of metabolic syndromes.

3. The authors characterize the hepatic epigenome by H3K27ac ChIP-seq, and discover approximately 4500 regions differentially occupied by H3K27ac in the two dietary conditions. The data quality is questionable, since the correlation between samples within the same dietary condition is not higher the correlation between samples across diets. It would be beneficial to explore data quality further by for example PCA.

We apologize for the confusing description. The overlap rate in the supplementary table was only calculated by the genome localization of the peak region, not the correlation. We measured the similarity of our sample by DiffBind using the default settings and present the data below (only presented here). The PCA analysis and correlation analysis use the ChIP-seq data at all binding regions. The samples were clearly clustered (figure a below) by the diet and each replicate had high correlation (figure b below) within condition. We also present the PCA and correlation analysis (figure c and d below) using only the differentially bound regions. It also shows that the samples are clustered well by diet. In order to find the high-confidence regions that changed by the diet, we only include the differential regions that occur in at least

two replicates in each diet in the further analysis.

4. The authors report that regions differentially occupied by H3K27ac are enriched in the vicinity of genes related to immune function, and speculate that this reflects the increased liver inflammation associated with NAFLD. This conclusion is not particularly convincing since the regions are not stratified by dietary condition. Thus, it is not clear whether the regions enriched for H3K27ac in the HFD condition are actually associated with inflammatory genes. The authors also perform motif analysis in the differentially occupied regions, but do not comment on the potential implications of their findings. Here, it would also be beneficial to stratify the regions by the diet in which they show enrichment of binding.

Thanks for the suggestion. We now stratify the differentially enriched H3K27ac regions by diet and did the pathway and motif analysis again. The regions enriched for H3K27ac in the LD condition were associated with inflammatory genes (figure S2d-g). We also added more

information about the potential implications of these data in the results.

5. The authors perform conventional Hi-C as well as promoter capture Hi-C but unfortunately these data are not really used in the manuscript to provide new biological insight. The authors show that the majority of A/B compartments as well as TADs do not change between the two diets. However, similar findings have been reported in several other systems, and as such it does not provide new insight. It could have been interesting to characterize the TADs or A/B compartments that do change, but unfortunately the authors did not do this.

Thanks for the suggestion. In our study, we found 5912 and 5776 TADs under LD and CD condition, respectively. As we presented in figure 2e, nearly 93% of TAD boundaries from each diet condition are within 20kb of a TAD boundary from the other diet. Another approach

is to assess how the TADs from each diet condition align against each other. Most TADs have a one-to-one pairing across diet conditions. The signal between these overlapped TADs are highly correlated (figure below a, only presented here). Another ~15% are cases where a TAD in one diet condition covers the same genomic span as two or more TADs in the other diet condition, like the example in middle of figure 2c. We refer to that as a "TAD skip" in the table below. In order to understand whether there is a true difference at the "TAD skip" regions, we divide the genomic space under those TADs into three categories: "one" (large TAD that contains more than one TAD in the other diet condition), "many" (smaller TADs that fall within one larger TAD in the other diet condition), and "outer" (genomic space that is within the "one" TAD but outside the "many" TADs) (see illustration in figure b below, only presented here) and compare their signal intensity. As shown in figure c (only presented here), the "outer" regions are as highly correlated as the shared TAD space, suggesting the presence/absence of a TAD boundary in such cases may be near the threshold. The remaining TADs have larger offset of their genomic positions (>50kb), but the relative signal between diet conditions is not sufficiently high that we have any confidence that those differences have biological meaning vs noise.

Description	LD		CD		Average
	Count	Percentage	Count	Percentage	
one-to-one TADs	4826	81.6%	4826	83.6%	82.6%
TAD skip	930	15.7%	801	13.9%	14.8%
Other	156	2.6%	149	2.6%	2.6%
Total	5912		5776		

6. The authors claim based on promoter capture Hi-C and H3K27ac ChIP-seq that HFD is associated mostly activation of enhancers that regulate genes across stable loops, while the high carbohydrate diet is associated with both enhancer activation and regulation of promoter interactions. This claim is not very convincing, since for promoter capture Hi-C, $N=2$ which indicates a very low statistical power to identify differential interactions in a sparse dataset. Furthermore, rigorous quality control and analysis of inter-replicate variance that may confound the number of up- and downregulated interactions are missing.

Thanks for this comment. Although we only have 2 replicates for capture Hi-C and 3 replicates for H3K27ac, our cutoff for identifying the differential H3K27ac binding sites and differential interactions was still conservative and acceptable for genome-wide studies (PMID:12883005). We calculated the percentage of the true null hypotheses and false discovery rate using the qvalue R package. The false discovery rate is around 7% (figures below, only presented here) in the significant differential interactions. We understand the reviewer's concerns about the over interpreting the data and we modified our conclusions in the results and discussion.

7. Using motif analysis in the promoters of differentially regulated genes, the authors identify HNF4A as a potential transcriptional regulator and map it using ChIP-seq in the two dietary conditions. Since the majority of HNF4A binding sites are stable between the dietary regimes, the authors speculate that HNF4A is not directly regulating dietary reprogramming, but serves as a bookmark for another transcription factor. Using motif analysis in HNF4A-bound enhancers, they find enrichment of the C/EBP motif and profile C/EBPa using ChIP-seq. The evidence that HNF4A bookmarks C/EBPa enhancers is based on co-occupancy analysis and lacks both statistical and experimental support.

Thanks for this comment. We agree that Hnf4a is labeling the active enhancers in liver cells and already added this in the text. Knocking out Hnf4a in mice liver will significantly decrease C/EBPa binding, which means Hnf4a is not only co-localized with C/EBPa, but also has functional effects on C/EBPa (PMID:23911320). Hnf4a is essential and whole body or liver-specific knockout of Hnf4a will cause death (PMID:7958910, PMID:11158324). Thus, it is hard to study long term metabolic adaptation using Hnf4a knockout mice. To understand Hnf4a's role on C/EBPa binding during the long term adaptation, we divided the diet-induced C/EBPa binding sites into two categories: sites pre-marked with Hnf4a or sites without Hnf4a. We found that diet-induced C/EBPa binding sites pre-marked with Hnf4a will significantly change the binding magnitude of C/EBPa binding in a diet-dependent manner (figure 6n).

While considering the loop with Hnf4a, we found it was different and loop connected Hnf4a increased C/EBPa binding in both diet conditions (figure 6o). These data support our hypothesis that Hnf4a might be functional as a bookmarker for C/EBPa during long term metabolic adaptation.

Specific points:

- Figure 1

o Figure 1D: The p-values should be adjusted for multiple testing.

Thanks for this comment. We used FDR value in the pathway analysis and they are now included in new figure 1e and f.

o Figure 1D: Are the upregulated genes in HFD associated with a different set of pathways compared to the downregulated genes?

Thanks for this comment. We did the pathway analysis under different diet conditions. The pathways enriched under different diet conditions were different (Figure 1e, 1f).

o Figure 1K: Are the upregulated genes in HFD associated with a different set of motifs compared to the downregulated genes? H3K27ac is a mark of the nucleosomes surrounding the nucleosome free region of enhancers. This nucleosome free region is the primary binding site of transcription factors to enhancers. Motif analysis should be redone to focus on these regions rather than the entire H3K27ac region.

Thanks for this comment. We now did the motif analysis under different diet conditions (figure 1k and l). The motif enriched in different diet conditions had some similarity and we also added more discussion in the results about the motifs.

• Figure 2

o Figure 2C/E + Figure S3B: There are clearly a significant number of regions that do change between diets. One example is in the center of the visualized regions, where two TADs merge. Are the genes that change between diets associated with these dynamic TADs or A/B compartment switches?

Thanks for the suggestion. As we described in the previous comments, the vast majority of the LD and CD TADs are very similar, and we do not have confidence that the limited number of TADs that do not closely overlap are meaningfully different. However, we did check the coordinates of the DEGs and found that 2.5% (93 of 3729) are located at non-overlapping (i.e. potentially 'different') TADs. A p-value for this enrichment was calculated as 0.897 by Monte Carlo random draw approach (i.e. 10,000 iterations of 3729 randomly-selected genes, excluding genes not expressed in either diet condition), indicating that there is no significant enrichment of DEGs at the potentially 'different' TADs.

• Figure 3

o Figure 3A: Capture efficiency for Hi-C is nonsensical, and cannot be used as a gauge for quality or enrichment.

We agree. We have altered the axis label accordingly to indicate that the Hi-C data is shown for comparison purposes to capture Hi C only, not as an intended quality assessment of the Hi-C data itself.

o Figure 3B/C: Have the Hi-C libraries and PHi-C libraries been subsampled to the same number of validations reads? Were significant interactions detected by the same method? We used iterative correction and eigenvector decomposition (ICE) normalization to normalize the Hi-C contact matrix rather than subsampled as described by Imakaev et al. 2012 Nat Methods (PMID: 22941365). As presented here, all replicates showed a high correlation and we combined them in the Hi-C to get higher resolution in each diet condition. The significant interactions in Hi-C data were identified by HICexplorer and described in the methods. For the capture Hi-C, we used CHiCAGO, which can use the replicates information in each group to call the significant interactions.

o Figure 3G (and manuscript lines 224-226): The authors state that even silent genes are enriched for interactions, which are present prior to enhancer activation. Can the authors comment on the number of false positives expected in their data? How many of these preformed loops connect to restriction fragment associated with enhancer marks in either diet or how many have the potential to be enhancers in different settings (e.g. in ENCODE data)?

We compared these interactions with silent genes under different diet conditions with ENCODE data using <https://epiexplorer.mpi-inf.mpg.de/>. In figure3 h, we found those loops were also highly overlapped with regulatory elements including DNase I activity sites, H3K4me1 and H3k27ac sites in ENCODE data comparing with size matched random regions, which means a large percentage of them have the potential to be enhancers.

o Figure 3H: Related to the above question concerning false discoveries. Does this trend persist if the analysis is restricted to very high confidence interactions (CHiCAGO score > 10)? The conclusion could be strengthened by showing that not only does gene expression associate with H3K27ac in an on/off manner, but also in a linear correlating relationship. Thanks for this comment. CHiCAGO score, based on the $-\log$ weighted p-values and p-value weighting procedure implemented in CHiCAGO, provides a multiple testing correction. The score threshold of 5 is a suggested stringent score threshold for calling significant interactions (PMID:27306882). CHiCAGO score 5 is default setting and also widely used in the capture Hi-C studies (PMID:26637943, 28475875, 29955040, 30305613, 29988018, 28332981). We agree that while using CHiCAGO score > 10 may reduce the false positive interactions, this is beyond our aim of current study. We also did spearman correlation (figures below, only presented here) between gene expression and number of interactions under different diet conditions. This showed that the number of interactions were significantly correlated with gene expression, which supports our conclusion.

• Figure 4

o Figure 4A/B (and manuscript lines 237-241): Please clarify if these tests have been adjusted for multiple testing and what the expected level of false discoveries is? What is the magnitude of change in either direction?

Thanks for the comments. As indicated figure a, we used qvalue R package to calculate the

portion of true null hypotheses of the interactions. The π_0 value is 0.704, which indicates that 70% of the interactions aren't different between the LD and CD groups. Our cutoff ($p < 0.001$) resulted in $\sim 1\%$ significant differential interactions, which is still conservative and acceptable for genome-wide studies (PMID:12883005). In our data, the p-value cutoff 0.001 is equivalent to q-value 0.07 (figures below, only presented here) and the expected level of false discoveries is around 0.07. We understand that multiple tests can cause false positive results, so in another Capture Hi-C study, other researchers used FDR 0.1 as the cutoff for significantly different interactions and FDR 0.7 as the cutoff for unchanged/stable interactions to avoid the false positive and false negative results (PMID: 28805829). We have added the magnitude of change in either direction to the text.

o Figure 4C: Which pathways are associated with interaction in each diet?

We have now stratified the differentially interactions by diet and did the pathway analysis again (now reported in figure 4c and d).

o Figure 4D: How many genes associated with LD-enriched interactions is also significantly higher expressed in LD compared to CD and vice versa?

We have included the DEGs with static loops as a control and are now reporting the number of DEGs associated with diet induced interactions in the results.

o Manuscript lines 265-268: The authors note that the number of significantly changing interactions is less than the number of significantly changing H3K27ac sites. For these numbers to make sense, it is important to consider the statistical power of the assays. The PChi-C dataset is likely to be sparse (which increases variation) and has N=2, while H3K27ac is likely more dense and has N=3. Thus, the conclusion here is confounded by differences in statistical power, and this needs to be taken into careful consideration.

Thanks for this comment. We understand the reviewer's concern about over-interpreting the data. We are more cautious now and have modified the conclusions in the results.

o Manuscript lines 268-274: Along the lines above, these conclusions might also be confounded by differences in statistical power, which needs to be carefully reviewed. An analysis of replicate similarity would be necessary. Furthermore, it would be sensible to subsample the libraries to a similar size. In the full dataset, the CD, which is associated with the majority of enriched contacts, is also sequenced approximately 40 million valid read pairs deeper per replicate. However, even with these updates, it is a difficult conclusion to reach

considering the low replication (which is common for these types of experiments, but a hindrance for exactly this type of conclusion).

Thanks for this comment. As described above, we applied ICE normalization to the Hi-C contact matrix rather than subsampling. As presented here, all replicates showed a high correlation (figure below) and we combined them in the Hi-C to get higher resolution in each diet condition. We understand the reviewer's concern and have modified the conclusions in the text.

o Figure 4G: How many genes associated with interactions with LD-enriched H3K27ac signal is also significantly higher expressed in LD compared to CD and vice versa?

We now added the number of DEGs in the text and also included the DEGs with unchanged H3K27ac signal as controls in figure 4h.

• Figure 5

o Figure 5A: These motif analyses should be performed in enhancers connected to the promoters of DEGs rather than the promoters of DEGs alone, and in a manner that separated out the individual diets.

Thanks for this comment. We want to understand the motif enriched in the DEG, not only the enhancer-connected DEGs. Since the promoter-enhancer interaction identified by Capture

Hi-C had a distance bias (it is difficult to detect the interaction within 50KB), we use the all DEGs as input to find the potential enriched motifs.

The authors speculate that HNF4a is activated by lipid-derived ligands in the lipid-rich diet regime, since there is an increased number of binding sites in this condition, although there is also less protein present. How is the nuclear abundance of HNF4a affected by diet? Both in total and relative to the amount of HNF4a present.

Hnf4a is localized primarily in the nucleus and immunoblot results indicated total Hnf4a was decreased under the LD condition. The increasing number of binding sites doesn't reflect the Hnf4a protein level. Lipids-rich diet contains lots of fatty acids, some of which might be the ligands for Hnf4a. Under the LD condition, Hnf4a was activated by its potential ligands to regulate gene expression.

Figure 5F: These analyses should be performed by using the PChI-C data rather than associating enhancers to genes by linear distance. Furthermore, it would be beneficial to choose an analysis and plot that is easier to interpret than the cumulative distribution.

Thanks for this comment. We regenerated the figure using both PChI-C data and also by assigning DEGs to differentially enriched Hnf4a binding sites using the linear distance.

In new figure 5i, we categorized the DEGs in 3 groups like figure 5h indicated: DEGs with loops connected to stable Hnf4a binding sites, diet induced Hnf4a binding sites, and no Hnf4a binding sites. We found that DEGs on the loops with diet induced Hnf4a binding sites had significantly higher expression than those on the loops with stable Hnf4a binding in both diet conditions. Due to the limitations of capture Hi-C, it had less power in capturing promoter enhancer interactions occurring over short (within 50 kb) distances (figure 3e). We assigned diet induced Hnf4a binding sites to the TSS +/-50kb of DEGs and also found that DEGs with diet induced Hnf4a binding had significantly higher expression than DEGs with unchanged Hnf4a binding. These data supported the idea that diet induced Hnf4a binding correlated with upregulation of gene expressions under long term adaptation.

• Figure 6

Figure 6A: The choice of control region is critical for this type of analysis, and randomized HindIII regions are a poor choice as this underestimates the background. In the literature

there are examples of other ways. Probably the best way is to shuffle the promoter labels of the detected interactions, as this preserves the structure of the data set but randomizes the location of the anchors in the genome. Another possible solution is for each bait-target pair to choose the distance-matched HindIII fragment in the opposite direction of the target as a control.

Thanks for this comment. Overlap of significant interactions with HNF4a peaks was assessed by the CHiCAGO function "peakEnrichment4Features", in which a proper control was evaluated including comparison to 100 random samples with equivalent distribution of interactions as recommend by CHiCAGO. We apologize for the misleading labels and revised the methods to avoid confusion by other readers.

o Figure 6B: Please include DEGs that do not show gain of HNF4a binding as controls.

Thanks for this comment. We included the DEGs with stable Hnf4a binding sites in fig 5i.

o Figure 6E: Is this overlap significant? This can be addressed by shuffling the binding sites of each factor randomly between enhancers and evaluating the randomized overlap.

We regenerated the Hnf4a and C/EBPa overlap in new figure 6c. The statistical significance of the overlap between HNF4a and C/EBPa peaks was determined by Monte Carlo simulation, where BEDtools v2.24.0 'shuffle' was used to select size-matched random peaks within enhancer regions (defined by H3K27ac peaks) followed by overlap assessment via BEDtools v2.24.0 'intersect' for 10,000 iterations. The p value was $p < 0.0001$. We have added this in the methods.

o The authors speculate that HNF4a bookmarks enhancers for C/EBPa, since it does not change in occupancy at the sites, where C/EBPa changes. The evidence present here is not sufficient to arrive at this conclusion, as there is no evidence to suggest any relationship between their binding other than co-occupancy. This claim should be backed by experimental evidence, e.g. HNF4a knockdown followed by C/EBPa ChIP(-seq).

Thanks for this comment. Knocking out Hnf4a in mice liver will significantly decrease C/EBPa binding, which means the Hnf4a was not only co-localized with C/EBPa, but also has functional effects on C/EBPa (PMID:23911320). The challenge to performing this experiment is that whole body knockout of Hnf4a is embryonic lethal (PMID:7958910) and mice with liver-specific knockout of Hnf4a will die around 8 weeks age (PMID:11158324). Thus, it's hard to study the long term (20weeks) metabolic adaptation using Hnf4a knockout mice. To understand Hnf4a's role on the C/EBPa binding during long term metabolic adaptation, we took our data and divided the diet-induced C/EBPa binding sites into two categories: sites pre-marked with Hnf4a and sites without Hnf4a (in figure 6n). We saw that diet-induced C/EBPa binding sites pre-marked with Hnf4a changed the binding magnitude of C/EBPa in a diet-dependent manner (figure 6n). While considering the Hnf4a binding on the loops, we found it was different and loop connected Hnf4a increased C/EBPa binding in both diet condition (fig 6o). These data, while not a perfect experiment, support our hypothesis that Hnf4a might be functional as a bookmarker for C/EBPa during long term metabolic adaptation.

Minor points:

- Overall

- o The nomenclature of diet regime should be consistent throughout the paper. The diets are referred to as CD/LD, HFD/LFD or control/NAFLD in a mixed manner.

Thanks for catching this. We revised the text carefully and now the diets are referred to as CD/LD.

- Figure 1

- o The understanding of the dataset could be improved by an introductory schematic figure of the experimental setup and readouts

We added an introductory schematic figure of the study in figure 1a.

- o Figure 1C mentions up- and downregulated genes, while the manuscript (lines 106-107) mentions genes upregulated in either of the diets.

We revised the text according to your comments.

- o The visual impressions of all screenshots (Figure 1F, H and J) can be improved.

We revised the figures according to your comments.

- Figure 2

- o The zoom-ins should be highlighted differently than dotted lines in Figure 2A/B. They are not easy to see or figure out that they demarcate regions of interest.

We revised figure 2a/b to make them easier to see.

o For consistency of analysis, it would be beneficial to show the length distributions of interactions in the same manner for Hi-C (Figure 2D) and PCHI-C (Figure 3E).

We revised fig2d and 3e to show the distribution in the same manner.

o Figure 2E: It might be easier to interpret this figure if simply the number or percentage of total boundaries that are conserved between diets and the number of boundaries that aren't is shown.

We added the number of boundaries that are conserved between diets.

• Figure 3

o Other papers have found similar findings as in Figure 3G. It would be nice to cite a few of these papers in the text to increase the confidence and accumulating evidence in the field that this relationship exist.

Thanks for this comment. We have cited some other works that have similar findings.

- Figure 4:

- o Figure 4E: The interactions in Figure 4E do not seem to be anchored in a promoter based on the annotation show in the figure.

We apologize for the misleading annotation and have revised the figure.

- Figure 5

- o Figure 5B: What is Control?

We apologize for the mislabel; "control" means CD group and we have revised the label.

- o Figure 5D: There is a spelling mistake on all x-axis labels in the figure.

We corrected the typos in fig5.

- o Figure 5E: Layer PCHi-C on top.

We added PCHi-C on top.

o Figure 5F: The tick marks should include 0.

We replaced figure 5f with new figure 5h-j.

• Figure 6

o Manuscript lines 346-348: The sites are enriched under a specific diet, not specific.

We added more details in the text and did the pathway and motif analysis on different diet conditions (figure 6g-j).

o Figure 6K (and manuscript lines 355-357): The authors use the screenshot as an example of their model of adaptation to diets. It would be helpful to have a schematic that highlights the important regulatory connections rather than a screenshot.

We made a schematic figure for our models (figure 6p).

Reviewers' comments:

Reviewer #1 (Remarks to the Author):

The authors have addressed all my previous concerns and their work merits to be published.

Reviewer #2 (Remarks to the Author):

The authors have implemented some of the proposed changes; however, I am generally not impressed by the revisions of the manuscript.

Regarding the original major points raised:

Re 1) The biological rationale for comparing the two extreme dietary challenges and not including a control remains unclear. What is the question? Where are the lean controls that was not exposed to high sucrose? How can one determine the effect of HFD/obesity when the control has been treated with high sucrose? The authors point out themselves that the term "low fat diet" is a misleading term when the diet has such a high concentration of sucrose and corn starch. I consider that a flaw in the setup, as this means that the authors with only two small groups of mice are comparing not just high and low-fat diet but also high and low sucrose. Furthermore, the fact that the physiological effects of these diets have been studied before doesn't free the authors from the need to determine whether the diets affect the mice as expected based on the literature. It is well known that different strains and substrains of mice respond differently to diets and that there is considerable variability also between individual mice of the same (sub)strain. I find it too superficial just to show weight increase and hepatic lipid accumulation. What are the effects of the high sucrose? Are the mice insulin resistant? What is the triglyceride content in the liver? Can inflammatory cells be detected in the liver?

Re 3) The plots in a and b should be included in the supplementary figures (c and d are non-informative as selecting differential sites pre-selects for sites that are similar between replicates and different between diets. Thus, it is not a good way to show the overall correlation). In b, the labels are misleading, as for example the light blue group in one diet is labeled CD1, while the light blue in the other diet is label LD2. Are the colors or labels the correct groupings?

Re 4) The authors have now stratified the H3K27ac regions by diet; however, from the text, it is not entirely clear whether the few motifs mentioned are exclusively enriched at diet-selective H3K27ac regions. The authors must compare the enrichment of motifs at diet-selective H3K27ac sites.

Re 6) For reasons explained in 1) the claim of responsiveness to carbohydrate diet and responsiveness to high fat diet seems impossible to make. Furthermore, the analysis does not provide strong support that H3K27ac signal is much more dynamic than PCHi-C interaction, since the power in H3K27ac seems to be greater than in the PCHi-C.

Re 7) The part on HNF4 remains rather weak, and this part of the manuscript would need considerable revisions and elimination of overstatements. The authors show that HNF4 α is expressed at lower mRNA and protein levels in HFD diet compared with high sucrose (whether it is the high fructose that enhance expression, or HFD that decrease expression remains unresolved). However, the comparison with the general observation that agonists increase the NR protein turnover is inappropriate for two reasons. First, it is unclear whether HFD leads to higher levels of HNF4 agonists; second, in the present work the authors are studying a long-term effect of diet and observe a decrease in mRNA expression, not just a decrease in protein expression. The finding that HNF4 α binding is increased in HFD when protein levels are decreased is puzzling and should be discussed. It is not correct to say that it is "consistent with the activation of HNF4 by lipid ligands.

The evidence that HNF4 α is a specific bookmark is lacking. The question is one that cannot be addressed by this type of study, because it is a long-term exposure to two very different diets.

There is no before and after, where one could claim that one factor was there before the other.

The authors could ask the question whether the two factors are more often bound to the same enhancer than expected by chance. That would give a hint as to whether the two factors bind in an

interdependent manner.

I agree that it would be difficult to do an in vivo loss of function experiment to demonstrate dependence between HNF4 and C/EBP binding, and I also don't think it is a sufficiently interesting question to justify difficult experiments. It is well known that many nuclear receptors cooperate with members of the C/EBP family in binding to chromatin.

Response to reviewers:

Reviewer #1 (Remarks to the Author):

The authors have addressed all my previous concerns and their work merits to be published.

We thank the reviewer for the constructive comments. We agree that the response to comments has improved our manuscript and thank the reviewer.

Reviewer #2

The authors have implemented some of the proposed changes; however, I am generally not impressed by the revisions of the manuscript.

We are sorry that the reviewer is not impressed with our response to review. We did our best to respond to the many comments (more than 40) provided.

Regarding the original major points raised:

Re 1) The biological rationale for comparing the two extreme dietary challenges and not including a control remains unclear. What is the question? Where are the lean controls that was not exposed to high sucrose? How can one determine the effect of HFD/obesity when the control has been treated with high sucrose? The authors point out themselves that the term “low fat diet” is a misleading term when the diet has such a high concentration of sucrose and corn starch. I consider that a flaw in the setup, as this means that the authors with only two small groups of mice are comparing not just high and low-fat diet but also high and low sucrose. Furthermore, the fact that the physiological effects of these diets have been studied before doesn't free the authors from the need to determine whether the diets affect the mice as expected based on the literature. It is well known that different strains and substrains of mice respond differently to diets and that there is considerable variability also between individual mice of the same (sub)strain. I find it too superficial just to show weight increase and hepatic lipid accumulation. What are the effects of the high sucrose? Are the mice insulin resistant? What is the triglyceride content in the liver? Can inflammatory cells be detected in the liver?

The question, we believe, is clear. As outlined in the abstract, introduction, results and discussion, we asked how long-range chromatin interactions differ in response to diet.

The reviewer asks for additional controls, citing the sucrose content of the carbohydrate rich diet. In fact, it is not possible to construct a diet comparison that alters only one variable. If we reduce calories from fat, we must replace them with something. The diet employed uses sucrose and corn starch to replace calories from fat. We could use an alternate diet, but this will also introduce an additional variable (increased protein content, increased carbohydrate content using something other than sucrose). We fail to see how ‘lean controls’ will permit isolation of fat (or carbohydrate) as a single variable. In addition, NIEHS normal mouse chow (which is NOT a defined diet) has 62% of calorie content from carbohydrate as compared to 70% in D12450B (although specific carbohydrates differ). It is unclear to us how adding animals on normal chow diet will simplify interpretation of the experiment rather than complicating interpretation with addition of one or more variables.

The reviewer also requests that we provide metabolic and physiologic characterization of the animals in our study. We concur that inclusion of such data would make our results more accessible to a general readership. We have performed a number of additional analysis of the animals in our study that are now depicted in the new Figure 1. We now have included animal weight (as a function of time), glucose and insulin challenge, resting levels of insulin and leptin, respiratory data and energy expenditure, lipid and cholesterol measurements, plasma liver enzyme levels to indicate liver damage, liver and adipose tissue mass and histologic analysis of liver and adipose. We agree that the inclusion of these data is important and we thank the reviewer and the editorial team for suggesting them.

Re 3) The plots in a and b should be included in the supplementary figures (c and d are non-informative as selecting differential sites pre-selects for sites that are similar between replicates and different between diets. Thus, it is not a good way to show the overall correlation). In b, the labels are misleading, as for example the light blue group in one diet is labeled CD1, while the light blue in the other diet is label LD2. Are the colors or labels the correct groupings?

We have included plots in a and b in supplementary figures. The figures were labelled correctly, colors in the replicate row simply indicate individual animals. We have modified the figure to eliminate any confusion.

Re 4) The authors have now stratified the H3K27ac regions by diet; however, from the text, it is not entirely clear whether the few motifs mentioned are exclusively enriched at diet-selective H3K27ac regions. The authors must compare the enrichment of motifs at diet-selective H3K27ac sites.

We performed the diet stratification requested in our previous revision. We label the new figures as CD induced and LD induced in the figure. In the text we state (line 166-169): "**In the lipid-rich diet enriched H3K27Ac regions**, the top enriched motifs corresponded to the known consensus binding sequences for ETS, bZIP and C/EBP...". Further we state (line 178-180): "**At carbohydrate-rich diet enriched loci**, we also found that motifs for the bZIP family transcription factors and nuclear receptor (HNF1, HNF6) families were highly enriched..."

We believe we have clearly identified the diet-selective regions as requested and performed the analysis requested.

Re 6) For reasons explained in 1) the claim of responsiveness to carbohydrate diet and responsiveness to high fat diet seems impossible to make. Furthermore, the analysis does not provide strong support that H3K27ac signal is much more dynamic than PCHi-C interaction, since the power in H3K27ac seems to be greater than in the PCHi-C.

We agree that the experiment performed in this manuscript is a bit more complex than simple response to diet. The animals in the study have vastly different metabolic and physiologic states. Assigning causality to a particular diet is likely simplistic. We have attempted to modify the manuscript to more accurately convey the reality that we are comparing animals that differ in multiple areas.

Re 7) The part on HNF4 remains rather weak, and this part of the manuscript would need considerable revisions and elimination of overstatements. The authors show that HNF4alpha is expressed at lower mRNA and protein levels in HFD diet compared with high sucrose (whether it is the high fructose that enhance expression, or HFD that decrease expression remains unresolved). However, the comparison with the general observation that agonists increase the NR protein turnover is inappropriate for two reasons. First, it is unclear whether HFD leads to higher levels of HNF4 agonists; second, in the present work the authors are studying a long-term effect of diet and observe a decrease in mRNA expression, not just a decrease in protein expression.

The reviewer expresses some lack of clarity on the presence of known ligands for HNF4a in the lipid rich diet. It is known that pork lard contains at least one known ligand for HNF4, linoleic acid (<https://doi.org/10.1371/journal.pone.0005609>) (typically approximately 10% by weight of lard is 18 carbon PUFA which includes linoleic acid).

It is certainly true that the current study is long term and that we observe both mRNA and protein level differences in HNF4a. We now discuss this in the text.

The finding that HNF4alpha binding is increased in HFD when protein levels are decreased is puzzling and should be discussed. It is not correct to say that it is “consistent with the activation of HNF4 by lipid ligands.

We do not agree with this comment by the reviewer. Hager and Yamamoto (PMID17261597) demonstrated that ligand binding by a nuclear receptor (GR) reduces its mobility in the nucleus by FRAP and increases its affinity for GREs. We now cite this relevant literature.

The evidence that HNF4alpha is a specific bookmark is lacking. The question is one that cannot be addressed by this type of study, because it is a long-term exposure to two very different diets. There is no before and after, where one could claim that one factor was there before the other. The authors could ask the question whether the two factors are more often bound to the same enhancer than expected by chance. That would give a hint as to whether the two factors bind in an interdependent manner.

We agree that the potential for HNF4 to bookmark sites for C/EBP is not directly tested in this study. We have altered the text accordingly and now refer to co-binding..

I agree that it would be difficult to do an in vivo loss of function experiment to demonstrate dependence between HNF4 and C/EBP binding, and I also don't think it is a sufficiently interesting question to justify difficult experiments. It is well known that many nuclear receptors cooperate with members of the C/EBP family in binding to chromatin.

We agree that this experiment, suggested by this reviewer, was not possible in our system.